# Vanishing weekly hydropeaking cycles in American and Canadian rivers

Stephen J. Déry [1✉], Marco A. Hernández-Henríquez [1], Tricia A. Stadnyk [2] & Tara J. Troy [3]

Sub-daily and weekly flow cycles termed 'hydropeaking' are common features in regulated rivers worldwide. Weekly flow periodicity arises from fluctuating electricity demand and production tied to socioeconomic activity, typically with higher consumption during weekdays followed by reductions on weekends. Here, we propose a weekly hydropeaking index to quantify the 1920–2019 intensity and prevalence of weekly hydropeaking cycles at 500 sites across the United States of America and Canada. A robust weekly hydropeaking signal exists at 1.8% of sites starting in 1920, peaking at 18.9% in 1963, and diminishing to 3.1% in 2019, marking a 21st century decline in weekly hydropeaking intensity. We propose this decline may be tied to recent, above-average precipitation, socioeconomic shifts, alternative energy production, and legislative and policy changes impacting water management in regulated systems. Vanishing weekly hydropeaking cycles may offset some of the prior deleterious ecohydrological impacts from hydropeaking in highly regulated rivers.

[1] Department of Geography, Earth and Environmental Sciences, University of Northern British Columbia, 3333 University Way, Prince George, BC V2N 4Z9, Canada. [2] Department of Geography, University of Calgary, Calgary, AB T2N 1N4, Canada. [3] Department of Civil Engineering, University of Victoria, Victoria, BC V8W 2Y2, Canada. ✉email: sdery@unbc.ca

In 2019, the United States of America (USA) and Canada generated a combined 674 TWh of hydroelectricity from a total 184 GW of installed capacity, ranking them with China and Brazil in the four largest global producers of hydroelectricity[1]. With the proliferation of dam and reservoir construction during the 20th and early 21st centuries[2,3], many of the two countries' main rivers are now moderately or strongly affected by fragmentation, regulation and/or diversions[4–6]. With increasing demands for renewable sources of energy, additional generating capacity is being developed or planned across Canada. This includes the 1,100 MW Site C Dam on the Peace River in northeastern British Columbia (BC), the 824 MW Muskrat Falls development on the lower Churchill River in Labrador, and the 695 MW Keeyask Generating Station on the Nelson River in northern Manitoba[1], with its first of seven units becoming operational in February 2021.

While overall demand for electricity continues to increase, consumption patterns vary depending on socioeconomic activity, short-term weather conditions, seasonal climate fluctuations and long-term climate trends[7,8]. In Canada, the winter season usually incurs peak electricity demand due to domestic, commercial and industrial heating and lighting requirements[9]. With climate change, winter cold waves subside while summer heat waves intensify[10,11], shifting some of the demand from winter heating to summer cooling[12–14]. Apart from seasonality shifts, day-to-day activities influence electricity demand as well. Similar to many other industrialized countries, North American educational, industrial and commercial activity intensifies on weekdays (Monday through Friday) but abates on weekends, particularly on Sundays[9]. This weekly rhythm of socioeconomic activity can thus impact water retention and releases in regulated rivers[15]. These rapid, frequent and periodic flow fluctuations downstream of regulation points are commonly termed 'hydropeaking' events and are known to disrupt a range of ecohydrological processes[16,17]. Yet, the characteristics and trends in weekly hydropeaking cycles due to daily variation in electricity demands remain largely unknown. This is despite the general availability of discharge data at a daily time scale and the distinct weekly rhythm of socioeconomic activity including hydropower production, and hence water releases in regulated waterways, which impact ecohydrological processes.

To address that knowledge gap and a demand for global attention to hydropeaking rivers[18], we assess here the prevalence of weekly hydropeaking cycles for 500 gauging sites along rivers of the USA and Canada spanning a wide range of basin characteristics, regulation, hydrological and climatic regimes. Specifically, we develop a scale-independent and dynamic weekly hydropeaking index (WHI) with both time and frequency domain terms, allowing quantification of weekly flow periodicity. As such, the WHI defines the prevalence and intensity of weekly periodicity in flows tied to hydropower production. We show that the WHI captures well the typical weekly rhythm observed in hydropeaking rivers, with low flows on weekends when hydropower demand wanes then high flows on weekdays when hydropower demand waxes. Application of the WHI to 1920-2019 time series of river discharge then provides evidence of vanishing weekly hydropeaking cycles in many regulated rivers of the USA and Canada with the 2010s comparable to the 1920s for hydropeaking prevalence. We propose that increased commercial and industrial activity on weekends, a shift towards other modes of energy production, policy changes altering water management practices, electrical grid interconnectivity and deregulation of electricity generation, plus a relatively wet decade in the 2010s across parts of the study area are likely contributing factors to waning weekly hydropeaking cycles. Thus this work is particularly relevant for long-term planning within the hydropower industry, power system operators and water resources managers.

## Results

**Study area.** The USA and Canada harbor abundant freshwater resources that include some of the world's largest rivers (by annual volumetric flows) including the Mississippi, St. Lawrence, Mackenzie, Ohio and Columbia rivers[19]. Many of these rivers and/or their tributaries have been impounded for hydropower generation, flood control, irrigation, potable water supply, navigation and recreation, leading to fragmented river networks and regulated flows[4,6]. Indeed, numerous dams have been built across the USA and Canada in the 20th and early 21st centuries[2,3]. Most dams in North America are operated for multi-purposes shaping seasonal and subseasonal patterns. Hydropower remains a principal component for sub-monthly variations along with flood control. Distinct weekly patterns mark hydropower production except perhaps at run-of-river facilities and those supplying industries continuously in operation such as aluminum smelters or pulp and paper mills[8,9]. As such, this study focuses on both regulated and unregulated waterways of the USA and Canada to explore the prevalence and intensity of weekly periodicity in discharge.

**Overall WHI statistics.** As defined, the WHI quantifies weekly periodicity in flows, with larger positive values indicating stronger weekly hydropeaking cycles and negative numbers its absence. The 1980–2019 mean, median, and standard deviation of WHI for the 500 sites reach 0.183, 0.056 and 1.121, respectively (Supplementary Table 1). Thirty-eight sites attain a mean annual WHI ≥ 2.0 for 1980–2019 with another 64 sites achieving WHI ≥ 1.0. A list of sites with the top ten ranking WHI values reveals their wide regional distribution with foci in the Chattahoochee, Colorado, Etowah, Great Lakes-St. Lawrence, Nelson, Smith and Wallenpaupack drainage basins (Supplementary Table 2), all of which are heavily dammed. The Smith River near Philpott claims the top WHI score of 3.783 while the Namakan River shows the lowest score of −3.168. Some highly regulated systems such as Manitoba's Burntwood River, which funnels water diverted from the Churchill River into the Nelson River, exhibit large negative WHI values (−1.884) as Notigi (the upstream point of regulation) is a control structure for a large reservoir operated in a longer term (e.g., seasonal) manner. Similarly, while several large dams impound the Missouri River, they are managed not only for hydropower production but also for flood control, irrigation, navigation and recreational values. As such, the four sites along the Missouri River used in this study exhibit an average WHI = −0.416 revealing an absence of significant weekly hydropeaking cycles.

**Spatial analyses.** A map of the 1980–2019 average annual WHI values reveals that weekly hydropeaking rivers abound across the USA and Canada. Clusters of high WHI values emerge in the Alabama, Chattahoochee, Cumberland and Tennessee river basins of the southeastern USA, in waterways draining the Ozark Mountains, the Colorado River and in northern Ontario rivers draining into the Great Lakes (Fig. 1). The Columbia River has several major points of regulation (WHI ≥ 1.5) from its headwaters in BC to its outlet in the Pacific Ocean. Highly hydropeaking sites (WHI ≥ 2.0) appear in both small (e.g., Alberta's Kananaskis River, $A = 899$ km$^2$) and large (Manitoba's Nelson River, $A = 1.1 \times 10^6$ km$^2$) systems. In contrast to their adjacent regulated rivers, free-flowing rivers of northern Canada, particularly those draining into Hudson Bay, exhibit large, negative

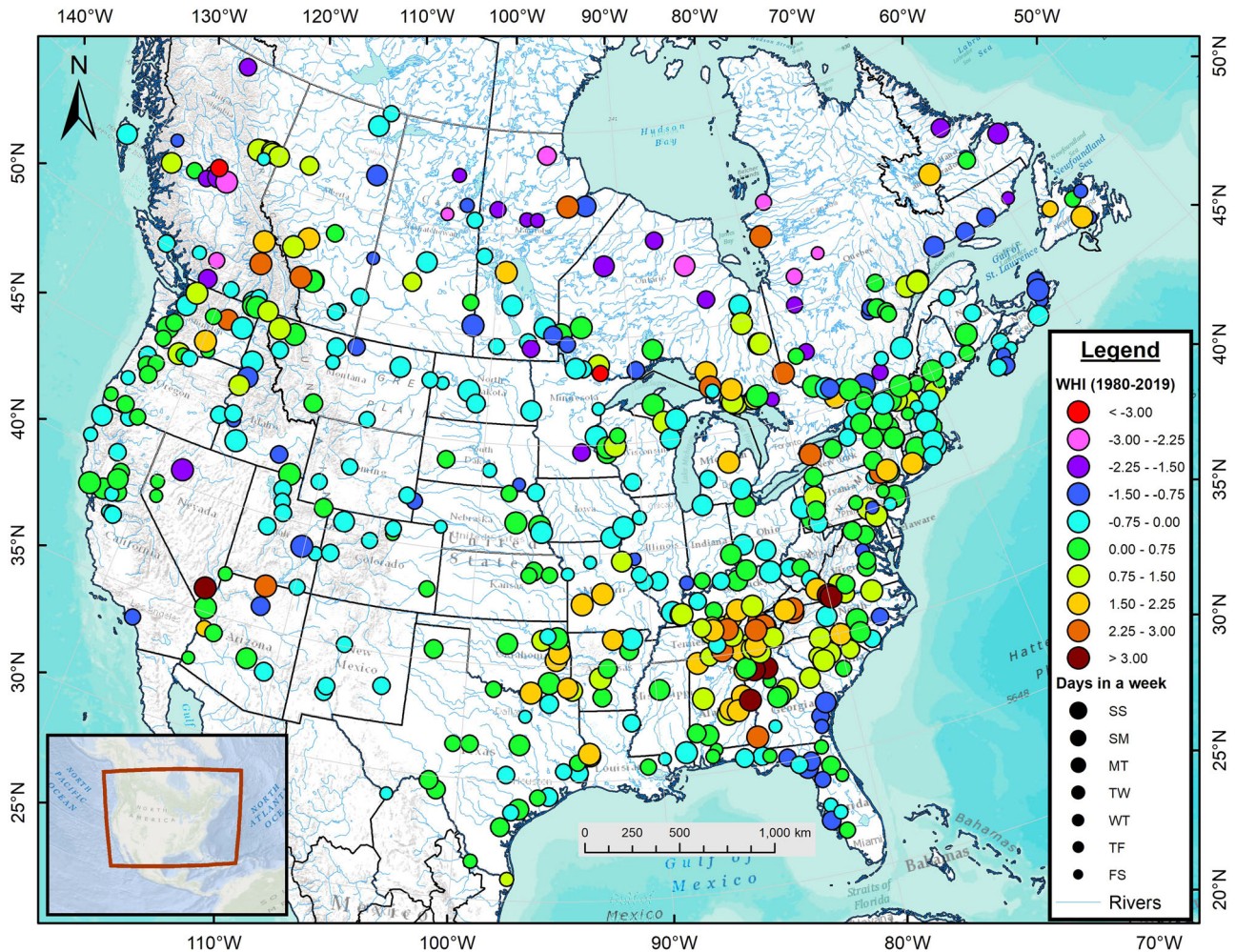

**Fig. 1 Map of the 1980–2019 mean WHI values for 500 sites across the USA and Canada.** Circle size corresponds to the two consecutive days with low flows beginning with the Saturday/Sunday (SS, largest symbols) combination and ending with the Friday/Saturday (FS, smallest symbols).

WHI values. These unregulated rivers manifest strong annual cycles dominated by snowmelt-driven freshets and contain large natural storage capacity in the form of extensive lakes, ponds and wetlands. Free-flowing, pluvial rivers of the southeastern USA (e.g., the Choctawhatchee, Ogeechee, Pascagoula, Satilla and Suwannee rivers) also exhibit negative, albeit $> -1.5$, WHI scores. More than 20% of Canadian rivers exhibit WHI $< -1.5$ while this proportion reaches 0.6% in American waterways (Supplementary Table 3). Nevertheless, both countries have a similar fraction of sites exhibiting WHI $> 0.75$ with 26.4% in the USA and 23.7% in Canada.

WHI values diminish moving downstream from a point of regulation. For instance, WHI $= 1.437$ on the Peace River just downstream of BC's WAC Bennett and Peace Canyon dams where minimum flows arise on weekends; 400 km downstream from the dams[20], however, WHI declines to 0.912 at the community of Peace River in Alberta where minimum flows occur on Mondays/Tuesdays, indicating a 2-day delay in signal propagation. A cascade of dams and reservoirs can accentuate the hydropeaking signals along waterways (e.g., the Colorado, Columbia, and Tennessee rivers) or attenuate them (e.g., Ottawa River).

Sites with high values of WHI ($\geq 1.5$) also show a preponderance of flow reductions on the weekends (Saturdays/Sundays) as identified by the larger symbols in Fig. 1. Of the 63 sites with WHI $\geq 1.5$, 56 experience the two consecutive days

with low flows on weekends. In contrast, sites with negative WHI values show a range of low flow days with no distinct pattern emerging. No less than 31.0% of all sites used in this study exhibit low flows on Saturdays/Sundays, more than twice the expected value (Fig. 2). This disproportionate amount of weekend low flows occurs mainly in hydropeaking rivers (WHI $> 0$). Weekday combinations show frequencies at, or lower than, the expected value with the Friday/Saturday sequence appearing at only 6.8% of sites. A Chi-Square test applied to the frequency of two consecutive low flow days reveals that the results differ significantly from the expected value of 0.143 ($\chi^2 = 136.33$, $p < 2.2 \times 10^{-16}$, $n = 7$ with six degrees of freedom). The mean WHI equals 0.326 for 155 sites with low flows on weekends while it remains near zero or slightly negative for the six other two-day combinations. The distribution of mean WHI for the two-day combinations differs significantly from a uniform distribution based on a Chi-Square test ($\chi^2 = 12.286$, $p = 0.027$ based on 10,000 replicates with $n = 7$).

**Temporal evolution and trend analysis.** The temporal evolution of the mean and median WHI shows a rapid increase in hydropeaking intensity from the 1920s to the 1950s at which point they level off and fluctuate near zero (Fig. 3). Starting in the 1990s, though, there is a gradual decline in both the mean and median WHI values with a return in the 2010s to statistics first seen in the

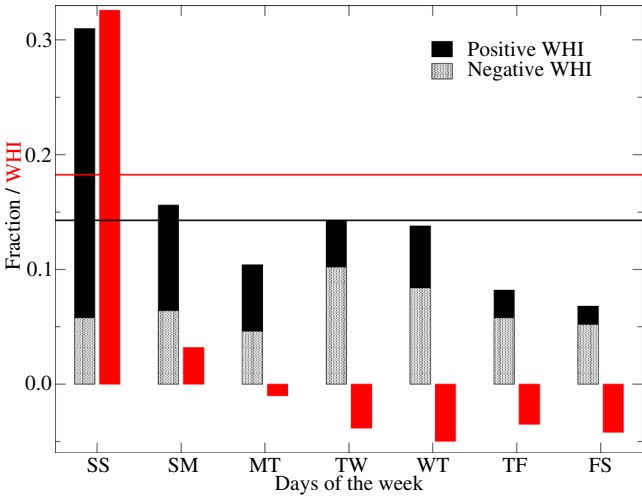

**Fig. 2 Histogram of the 1980–2019 frequency distribution of low flow days and corresponding WHI values.** Black bars denote the two consecutive days with low flows while red bars represent the WHI values for 500 sites across the USA and Canada, 1980–2019. Fractions of the two consecutive days with low flows are partitioned according to positive (solid) and negative (dotted) WHI values. The days of the week begin with the Saturday/Sunday (SS) combination and end with the Friday/Saturday (FS) combination. The horizontal black line denotes the expected value if the two-day low flows were distributed randomly while the horizontal red line marks the mean WHI across the 500 sites.

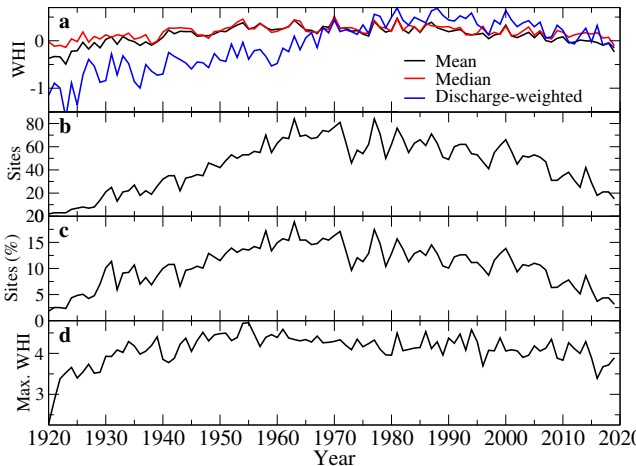

**Fig. 3 Temporal evolution of WHI metrics for 500 sites across the USA and Canada, 1920–2019.** The (**a**) annual mean, median and discharge-weighted WHI values for up to 500 sites in the USA and Canada, the (**b**) number and (**c**) percentage of available sites ranking in the top decile of all WHI values, and the (**d**) annual maximum WHI values, 1920–2019.

1930s (largely pre-regulation), a pattern observed both in the USA and Canada (not shown). The discharge-weighted $WHI_Q$, whereby the sum of WHI times mean annual discharge are then normalized by annual discharge summed at all available sites, emphasizes the increasing volumes of regulated flows starting from the 1920s through the 1980s; however, $WHI_Q$ also declines markedly thereafter into the 21st century. In 1920, only 1.8% of available sites rank in the top decile of 1920–2019 WHI values (WHI ≥ 2.117). This fraction peaks at 18.9% of available sites in 1963 but thereafter diminishes consistently. In 2000, 66 or 13.8% of available sites score in the top decile of 1920–2019 WHI values but these counts fall precipitously to just 15 or 3.1% of the

available sites by 2019, marking a 21st century declining pattern in weekly hydropeaking intensity. Trend analysis applied to the overall mean annual WHI reveals a statistically significant decline of −0.40 over 1980–2019 (Supplementary Fig. 1). These temporal results, however, rely on the availability of discharge data, as the record length averages 79.7 years, ranging from a minimum of 24 years at one site to a full century at 69 sites (Supplementary Fig. 2). The number of available sites increases steadily from 1920 into the early 1990s and peaks at 492 sites in 1985 and 1992 but then declines to 468 sites by 1996 thereafter averaging 483 ± 5 sites until 2019. Notable gaps appear in the discharge records starting in the 1990s, particularly for regulated rivers in Ontario and Québec; however, adjusting the time series of mean annual WHI for unavailable sites reveals little difference in the overall pattern and trend of WHI during 1980–2019 (Supplementary Fig. 1).

Data availability also factors in the appraisal of the decadal evolution of hydropeaking intensity across the USA and Canada (Fig. 4). Nevertheless, this shows the gradual inception of hydropeaking cycles during the 1920s and 1930s, particularly in the north-central, northeastern, and southeastern USA and in northern Ontario. The 1940s show an expansion of weekly hydropeaking rivers into the western USA including within the Colorado, Columbia and Sacramento river basins as the 1930s New Deal projects came online. The 1940s and 1950s mark an intensification of regulation in the Tennessee and Alabama river basins, the Ottawa Valley as well as rivers of northern Ontario draining to Lakes Superior and Huron. A pronounced expansion and amplification of the hydropeaking signal appears in the 1960s, particularly across the Great Lakes-St. Lawrence river basin in Ontario and Québec. Some stabilization of the hydropeaking pattern marks the 1970s but a resurgence follows in the 1980s and 1990s when additional hydropeaking rivers emerge in western Canada. The 2000s retain a wide distribution of hydropeaking rivers across both countries; yet, by the 2010s, the number of highly hydropeaking rivers diminishes considerably, particularly in parts of the Great Lakes-St. Lawrence and Tennessee river basins.

The decadal distribution of the 10 WHI bins (Fig. 5a) further highlights the peak fraction of sites with WHI ≥ 1.5 attained in the 1960s (20.7%), with nearly matching minimum values in the 1920s (6.8%) and 2010s (7.8%). After the 1960s, there is a steady decline in the relative number of sites with low flows either on the Saturday/Sunday or Sunday/Monday combinations, indicating waning differences between weekday and weekend flows across the USA and Canada (Fig. 5b).

The temporal evolution of the annual maximum WHI value shows a rapid increase from ~3.0 in the 1920s to > 4.0 in the 1930s onward (Fig. 3d). Annual peak WHI values > 4.0 are generally sustained for the remainder of the 20th century but then remain near 4.0 or below that threshold starting in 2003 until 2019. The peak WHI value each year over the study period is distributed among 21 sites, with the Smith River at Philpott capturing the top spot most often at 28 times and scoring the overall maximum WHI of 4.752 in 1955 (Supplementary Fig. 3).

Further statistical analysis reveals an abundance of strong, negative WHI trends interspersed with positive ones for the 479 sites with $n_y$ ≥ 30 years over 1980–2019 (Fig. 6). A total of 138 sites show locally statistically significant ($p < 0.05$) declines in WHI while 28 show locally statistically significant inclines. Of the 166 locally significant trends, 129 remain globally significant. Significant negative WHI trends abound in the southeastern and northeastern USA, the Great Lakes-St. Lawrence basin, and the Pacific Northwest while a cluster of positive trends arises in Québec's Saguenay watershed. Clusters of negative WHI trends lie primarily within the Western, Northeastern and Southeastern

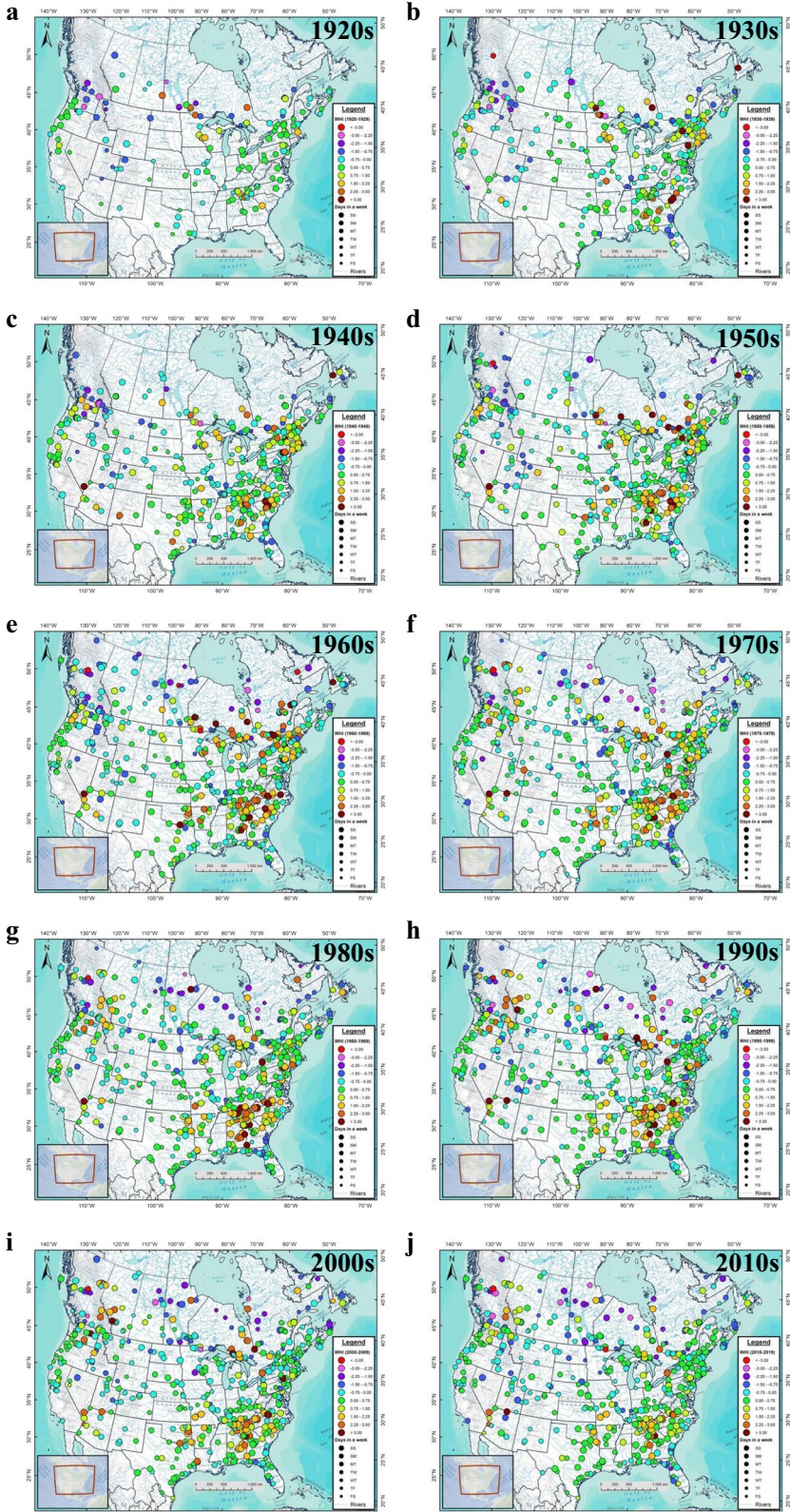

**Fig. 4 Maps of the decadal mean WHI values for 500 sites across the USA and Canada.** Maps are shown for (**a**) 1920–1929, (**b**) 1930–1939, (**c**) 1940–1949, (**d**) 1950–1959, (**e**) 1960–1969, (**f**) 1970–1979, (**g**) 1980–1989, (**h**) 1990–1999, (**i**) 2000–2009, and (**j**) 2010–2019. Circle size corresponds to the two consecutive days with low flows beginning with the Saturday/Sunday (SS, largest symbols) combination and ending with the Friday/Saturday (FS, smallest symbols). Results are shown only when $n_y \geq 5$ years in a given decade.

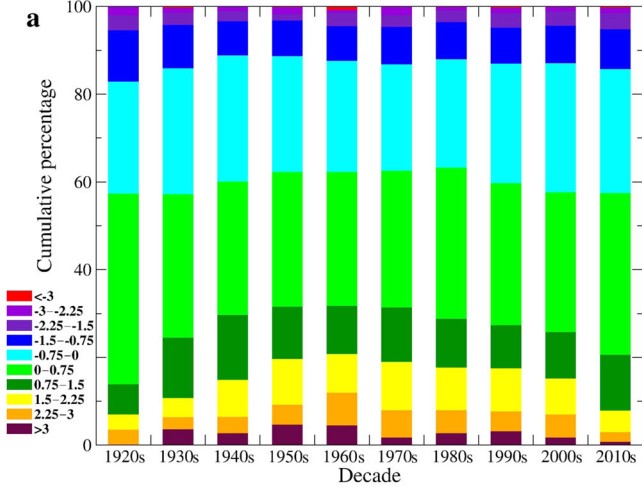

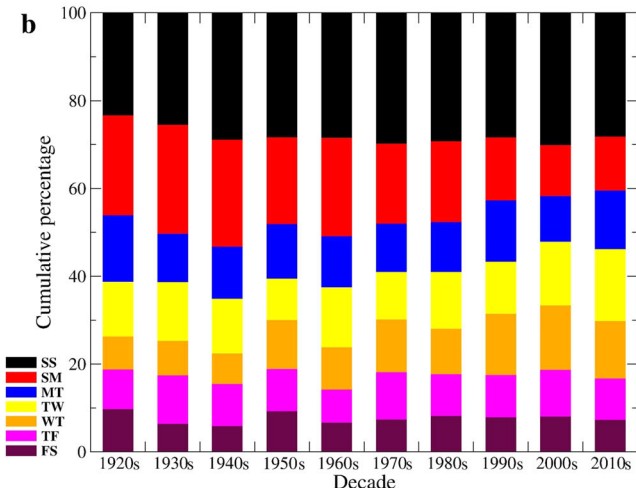

**Fig. 5 Decadal cumulative percentage of sites falling within one of 10 WHI bins and one of seven two-day combinations of low flows, respectively, across the USA and Canada.** In (**a**), WHI bins match those used in Fig. 4 with a similar color palette (e.g., the maroon bars indicate WHI ≥ 3.0 starting at a zero cumulative percentage). In (**b**), the two-day combinations with low flows start on Friday/Saturday (FS) at a zero cumulative percentage (maroon bars) and end on Saturday/Sunday (SS) at 100% (black bars).

Interconnects. While regulated rivers of Newfoundland show increasing WHI values, their unregulated counterparts show similar tendencies. Furthermore, in New Brunswick, the regulated Saint John River shows a decreasing trend in WHI while the proximal, unregulated Southwest Miramichi River shows an increasing trend. Sixty-nine percent of the locally significant WHI trends arise in hydropeaking rivers (WHI > 0) with fewer locally significant trends in non-hydropeaking rivers (WHI < 0; Supplementary Fig. 4). Application of the Pettitt test[21] reveals that two-thirds of the locally significant trends detected with the Mann-Kendall test also correspond to statistically significant break points in the WHI time series (Supplementary Data 1).

**Influence of dams and reservoirs**. The commissioning and operation of hydroelectric facilities including dams and reservoirs markedly influences WHI evolution. For instance, development of hydropower dams by the Tennessee Valley Authority (TVA) in the first half of the 20th century across the Tennessee River Basin induces sharp increases in WHI, often from large negative to positive values (Supplementary Fig. 5). In some cases, however,

the WHI is already elevated at gauging sites, reflecting the presence of additional upstream points of regulation (e.g., the Hiwassee Dam on the Hiwassee River commissioned in 1940). Consistent with the general pattern observed across the USA and Canada, most sites operated by the TVA show an attenuation of WHI values in the 2010s. Where present, reservoir influence on WHI depends on its function (Supplementary Fig. 6). When managed for hydropower production among other functions, the WHI typically stays positive with many surpassing scores of 2.0. At sites where reservoirs serve other purposes, the WHI remains substantially lower alternating between positive and negative values.

**Interannual and interdecadal variability**. Water management practices and climate variability, among other factors, yield significant interannual variation in hydropeaking intensity. For example, the Colorado River at Lees Ferry shows marked declines in WHI during high flow years (Supplementary Fig. 7a). Indeed, heavy precipitation during strong El Niño events in the early 1980s induced high flows in the Colorado River including at Lees Ferry. Due to the unusually wet weather, the bypass tubes and spillway at Glen Canyon Dam were used to release additional water downstream, thereby moderating hydropeaking signals from 1983 to 1986[22]. Similar declines in WHI appear in 1997 and 2011 when flows exceed the recent annual average. Computing the Pearson correlation coefficient between the 1980–2019 annual river discharge and the corresponding WHI yields 94 statistically significant negative correlations and only 19 statistically significant positive correlations (Supplementary Fig. 7b). Thus high flows over extended periods attenuate weekly periodicity even in heavily regulated rivers such as the Colorado.

This analysis suggests that sustained wet periods may attenuate hydropeaking intensity while dry periods may accentuate it. Binned distributions of decadal standardized anomalies in river discharge reveal the contrasting dry 1930s vs. the wet 1970s, the latter coinciding with a suppression of hydropeaking across the USA and Canada (Supplementary Fig. 8a). Yet, while the 2010s experienced relatively high flows, 7.8% of sites have WHI ≥ 1.5 whereas in the similarly wet 1990s, 17.4% of sites achieve WHI ≥ 1.5. Of 19 sites with large (> 1), positive standardized discharge anomalies during the 2010s, only four (the Betsiamites, La Grande and Nelson rivers plus Wallenpaupack Creek) have WHI > 1, which are likely more in response to enhanced diverted flows (excluding Wallenpaupack Creek) rather than high precipitation. While there are robust positive discharge anomalies in the north-central plains and northeastern USA and parts of central Canada in the 2010s, other regions with significant WHI declines exhibit near neutral or even negative discharge anomalies (Supplementary Fig. 8b). Thus it is unlikely interdecadal climate variations alone account for recent WHI declines.

**Dispersion of daily flows**. Apart from climate variations, changes in day-of-the-week flows may influence WHI trends. Sites with WHI > 0 generally observe greater dispersion of day-of-the-week flows although pluvial and intermittent rivers, particularly in the southern USA, also experience greater day-to-day flow variations (Supplementary Fig. 9a). A trend analysis reveals significant declines in the dispersion of flows across the seven days of the week, concomitant with diminishing WHI values from 1980 to 2019 (Supplementary Fig. 9b). As an example, an abrupt reduction in dispersion of day-of-the-week flows in Labrador's Churchill River appears in 1997 and is then sustained, suggesting factors other than climate variations are altering daily flows (Supplementary Fig. 10).

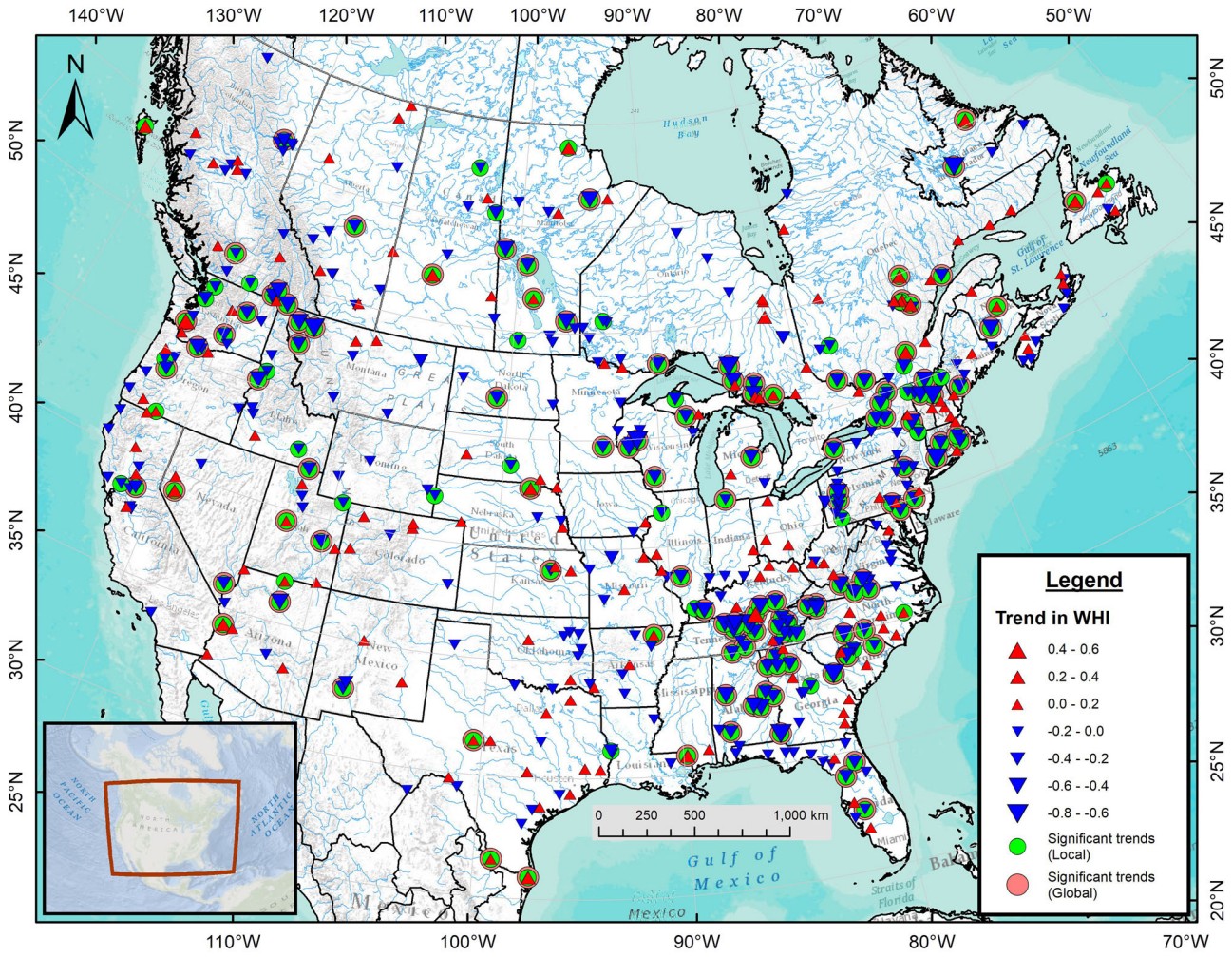

**Fig. 6 Map of the 1980–2019 monotonic trends in WHI at 479 sites across the USA and Canada.** Red upward (blue downward) pointing triangles indicate positive (negative) trends. Trend magnitudes are proportional to the triangle sizes and green circles (pink outlines) indicate locally (globally) statistically significant trends ($p < 0.05$). Results are shown only when $n_y \geq 30$ years.

## Discussion

**Possible factors leading to recent WHI declines**. The recent decline in weekly hydropeaking cycles in the USA and Canada emerges as a key finding in this study. Several possible factors may be contributing to this general pattern observed over the study area. Firstly, electricity demand, production and consumption may have shifted in recent years, thereby diminishing differences between weekdays vs. weekends. For instance, there has been a gradual shift towards more commercial (including e-commerce) and industrial activity on weekends that could alter the weekly discharge patterns in regulated rivers[23,24]. A shifting manufacturing sector, globalization, and lifestyle changes are all socioeconomic factors modifying electricity demand. Another possible factor is the development and expansion of other modes of energy production such as dispatchable combustion turbines and non-dispatchable solar and wind energy (Supplementary Fig. 11). Solar and wind energy production activate during favourable weather conditions with hydropower otherwise matching the demand, which may disrupt the typical weekly pattern in regulated flows while allowing hydropower to offer new types of services such as capacity markets. Furthermore, the rapid increase in electricity production from non-hydro renewable sources coincides with the sharp decline of weekly hydropeaking intensity in the 2010s (Supplementary Fig. 12).

Regulatory bodies and changing governmental policies may also be altering how utilities manage regulated waterways. Indeed, there is renewed interest for environmental, ecological and cultural (e.g., from a First Nations or Indigenous perspective) flows in human-influenced systems, with emerging regulations and policies supporting their implementation[25]. For instance, regulatory changes in the operation of the Prickett hydroelectric facility from a peaking to run-of-river site to assist spawning lake sturgeon[26] induced a significant WHI decline (of $-0.216$ decade$^{-1}$) along the Sturgeon River in the upper peninsula of Michigan starting in the 1990s. Indeed, changes in operation away from peaking hydropower generating stations, whether mandated or voluntary, could influence hydropeaking patterns.

The increasing interconnectivity of the North American power grid, deregulation, and centralization of electricity dispatching may further contribute to a recent reduction of hydropeaking intensity. Finally, climate variations may also play a role in hydropower production as wet periods may require greater spillage of water from reservoirs thereby diminishing hydropeaking intensity. Alternatively, wet years may lead utilities to generate continuous baseload energy instead of peaking hydropower, inducing a similar effect. The relatively wet climate of the 2010s could account for part of the recent declines in WHI across Canada and the northern half of the conterminous USA. Thus a

combination of factors including changing electricity demand patterns tied to lifestyle factors and socioeconomic activity, the emergence of alternative modes of energy production plus power grid interconnectivity, implementation of regulations and policies, and climate variations may be influencing the day-to-day hydrology of many regulated waterways across the USA and Canada.

**Spatio-temporal patterns within and across jurisdictions.** Given the vast territory of the USA and Canada, their waterways often drain multiple jurisdictions including international transboundary watersheds (e.g., the Rio Grande, Great Lakes-St. Lawrence, Winnipeg and Columbia rivers). Regional water authorities, inter-jurisdictional water boards, federal, provincial, and state legislation, and international water treaties and commissions all affect how waterways are managed. Furthermore, synchronous inter-jurisdictional power grids (e.g., interconnections) can also affect hydropower generation and hence regulated flows, leading to distinct spatio-temporal patterns in hydropeaking intensity. Decadal maps of WHI values reveal the progression of weekly hydropeaking systems from the eastern and central USA to the Pacific Northwest in the 1960s when development in the Columbia River Basin expanded rapidly. The international Columbia River Treaty implemented in 1961 led to the construction of three major dams along the Columbia River (Duncan, Keenleyside and Mica Dams in Canada) plus another on the Kootenai River (Libby Dam in the USA)[27]. These dams and generating stations expanded the presence of hydropeaking cycles from the lower to the upper Columbia River Basin in the 1970s and 1980s (Fig. 4). As such, regulation in the Canadian portion of the Columbia River Basin now leads to downstream propagation of hydropeaking into the northern USA where it is regenerated at multiple points of regulation including Grand Coulee Dam and the Dalles.

Another noticeable pattern in the decadal results is the WHI decline in many rivers of southern Québec in the 1970s and 1980s. As the 5,428 MW Churchill Falls generating station in Labrador came online in late 1971 (with hydropower sold mainly to the provincial utility Hydro-Québec)[28], followed a decade later by the 17,418 MW James Bay Hydroelectric Complex in northern Québec[15], a northward shift in hydropower generation abated the weekly hydropeaking cycles in more southern waterways. Simultaneous reductions in WHI in the northeastern USA (e.g., Hudson and Connecticut Rivers) may also be tied to transboundary power grid interconnections and Hydro-Québec's large export capacity (7,974 MW in 2019[29]). Similar to regional climate trends[30], synchronous power grids thus have the capacity to shift the intensity of hydropeaking signals 1000s of kms away from points where hydropower is consumed, thereby creating *hydropeaking teleconnections* with potential for far-reaching social and ecohydrological effects.

**Ecohydrological implications.** Ecohydrological impacts of hydropeaking are site-specific and may include rapid changes in water temperature (i.e., 'thermo-peaking'), increases in soil erosion and suspended matter, and habitat degradation, which affect ecosystems, reduce species abundance, and limit biodiversity (e.g., fish, riparian plants, macroinvertebrates)[16,31,32]. Across the USA and southern Canada, hydropeaking emerged relatively early in the 20th century with the proliferation of dams and flow regulation in these regions. Starting in the 1960s, hydropower infrastructure expanded northwards into regions previously devoid of any significant flow regulation and hydropeaking. This includes major waterways like BC's Peace River, Manitoba's Nelson River, Ontario's Moose and Abitibi rivers, and Québec's

La Grande Rivière. On these systems, major dams and reservoirs were built from the 1960s to early 1980s, vastly expanding the northern reach of hydropeaking rivers (Supplementary Fig. 13). This shifted potential ecohydrological impacts of hydropeaking to areas also undergoing rapid climate change through Arctic amplification of global warming[33]. As such, sub-Arctic species of fish (e.g., brook trout, lake sturgeon, northern pike, and walleye), insects and riparian plants may now be exposed to the cumulative impacts of these environmental stressors[17]. Additionally, winter frazil ice production and ice jams may be precipitated and accentuated downstream of hydroelectric facilities with persistent hydropeaking signals such as in the Peace River[20].

Despite their recent northward expansion, weekly hydropeaking cycles are generally waning across the USA and southern Canada, suggesting a 21st century *hydropeaking recovery* in some of these river systems. Indeed, prior ecohydrological impacts of hydropeaking may be partially offset, benefiting local biota and ecosystem biodiversity[34]. For instance, recovery of lake sturgeon in the northern peninsula of Michigan demonstrates some of the benefits of shifting away from peaking hydropower operations[26]. This is particularly important as evidence is also mounting that hydropeaking influences aquatic species in rivers of Canada[35–38]. Other aspects of flow regulation, such as sub-daily flow fluctuations and associated ramping up and down cycles not investigated in this study, may negate this hydropeaking recovery[16,17]. Additional research is thus needed to explore hydropeaking cycles at other temporal scales to establish their site-specific ecohydrological impacts.

**Advantages and limitations of the WHI relative to other metrics.** The proposed index to infer weekly hydropeaking signals provides a complementary metric to those developed in other studies[5,39,40]. Advantages of our approach include its scale independence, dynamic response, and relatively simple implementation. The WHI can be applied from small ($<1 \times 10^3$ km$^2$) to large ($>1 \times 10^6$ km$^2$) river basins with available daily discharge data (whether observed, reconstructed or simulated). The WHI responds to interannual variability in climate (e.g., wet/dry periods), changes in water management practices and policies, commissioning of new hydroelectric facilities or decommissioning of old ones, and other factors that affect flows. The use of daily discharge data also avoids the need for extensive databases on dams, reservoirs and other infrastructure that influence flows. Its possible implementation for short-term flow predictions emerges as another distinct advantage of the WHI. As an example, a running value of the WHI can be computed on the past year's daily flows and used to infer the possible deviations in daily flows over a given week based on recent historical patterns. Its computational simplicity, coded in our study in Fortran, allows processing of results for the 500 sites in <4 min. As such, it is feasible to implement a version of the code for short-term flow predictions so long as up-to-date daily flow records remain available. It would also be relatively straightforward to adapt the code to explore sub-daily hydropeaking cycles[9] if appropriate discharge data are available.

One challenge in implementing the WHI is access to daily discharge records. While considerable gauging stations exist in most of the USA and southern Canada, other waterways are not necessarily well monitored. A late 20th century decline in hydrometric stations due to budget restraints[41] and the Water Survey of Canada's curtailment of data collection combined with stricter quality standards from third parties have exacerbated hydrological data accessibility. As well, private industry and government-owned corporations often record discharge at or near their hydroelectric facilities, but may consider these data as

sensitive such that they are not released publicly or remain difficult to access. Thus, acquisition of daily discharge data in regulated systems, particularly as the number of small, private firms operating run-of-river hydroelectric facilities expands[3], yields a distinct challenge in accessing flow data. Therefore, remote sensing[42], data reconstructions (e.g., from statistical models or machine learning methods[43]) and numerical simulations that incorporate regulation[44] are key in filling spatio-temporal gaps where and when in situ observations are lacking.

**Summary and synthesis**. As hydropower generation and infrastructure development continue to expand across the USA and Canada, it is imperative to establish how water management practices affect downstream river flows and ecosystems. Common features in regulated rivers are discharge periodicities associated with hydropower production ebbs and flows including weekly cycles. In this study, a measure of this weekly rhythm in flows, the weekly hydropeaking index (WHI), is formulated and applied to 500 sites over parts of North America. Our analyses reveal that 29% of sites with at least three decades of available data during 1980–2019 exhibit locally statistically significant declines in WHI while only 6% show inclines. Moreover, the fraction of sites with WHI ≥ 1.5 dropped by half from the 2000s to the 2010s reverting to a value observed in the 1920s. Major watersheds observing significant declines in weekly hydropeaking include the Alabama, Columbia, Cumberland, Great Lakes-St. Lawrence, and upper Mississippi, which fall within the Eastern and Western Interconnects. Regional clusters of declining WHI highlight hydropower operations and river regulation governed at the watershed-, interconnect- and utility-scale.

Factors possibly yielding vanishing weekly hydropeaking cycles include increased commercial and industrial activity on weekends, a shift towards other modes of energy production during peak demand hours or days, and policy changes altering water management practices including for cultural, ecological and environmental flows. This reduction in weekly hydropeaking also may benefit aquatic species, insects and riparian vegetation that otherwise are susceptible to rapid shifts in flows and water levels. Future efforts should therefore establish the ecohydrological implications of waning weekly hydropeaking cycles. The application of the WHI to other regions over the globe would provide broader perspectives on the commonality of this feature in regulated rivers. Lastly, detailed investigations at various spatial (e.g., watershed, interconnect, utility) and temporal (e.g., seasonal) scales should be undertaken to elucidate the role of governing agencies and hydroclimate on hydropeaking globally.

## Methods

**Site selection**. A total of 500 sites across the USA and Canada ranging 190–1,805,222 km$^2$ in gauged area ($A$), 25–60°N in latitude, 54–132°W in longitude, and 0.02–268.28 km$^3$ in mean annual discharge are selected for this study (Supplementary Fig. 14 and Supplementary Data 2). A primary site selection criterion is daily discharge data availability for ≥24 years between 1920–2019, with ≥14 years during the focus period of 1980–2019. The chosen sites span a wide range of hydrological regimes from pluvial rivers in warmer climates (e.g., BC's Yakoun River) to nival and glacial systems at higher elevations or latitudes in cooler climates (e.g., BC's Lillooet River)[45]. Thus, the study area spans regions with little to no snowmelt where sub-annual scales govern temporal variability while others are mainly snowmelt-driven with predominant annual cycles[46]. The database also includes intermittent streams in warmer, drier climates such as California's Santa Ana River and Arizona's Little Colorado River. Regulated and unregulated rivers are selected (using guidance from Benke and Cushing[19]) to allow comparisons between sites. Some sites such as Lees Ferry on the Colorado River include extended records that cover pre- and post-regulation effects on flows.

**Data**. Data and metadata (station ID, gauge coordinates, and gauged area) are extracted from various sources including publicly accessible databases maintained by federal, provincial and state agencies in addition to proprietary or unpublished data from private industry, government-owned utilities and international

commissions. For most unregulated rivers, daily discharge data are sourced partly from the Water Survey of Canada's Hydrometric Database (HYDAT), the Centre d'Expertise Hydrique du Québec (CEHQ) and the United States Geological Survey (USGS). For regulated rivers, though, daily discharge data are not necessarily available from these sources or other public repositories as they are partially or entirely collected, quality controlled and archived by government-controlled utilities or private industry (see Supplementary Data 2 and 3). This includes: Nalcor Energy for the Salmon and Exploits rivers plus the Churchill Falls (Labrador) Corporation Limited for the Churchill River at Churchill Falls Powerhouse in Newfoundland and Labrador; NB Power for the Saint John River in New Brunswick; Rio Tinto for the Kemano Powerhouse in BC and the Saguenay and Péribonca rivers in Québec; Hydro-Québec for La Grande Rivière, Betsiamites, Gatineau, Manicouagan, des Outaouais, des Outardes and St-Maurice rivers; Evolugen by Brookfield Renewable for the Coulonge, Lièvre, and Noire rivers in Québec and Mississagi and Aux Sables rivers in Ontario; Ontario Power Generation for the Abitibi, English, Kaministiquia, Madawaska, Mattagami (tributary to the Moose River), Montreal and Ottawa rivers; H2O Power for the Abitibi River; Manitoba Hydro for the Nelson and Winnipeg rivers; TransAlta for the North Saskatchewan and Kananaskis rivers; and BC Hydro for the Columbia River at Mica Dam. Additional data for gauges along the Rio Grande on the border between the USA and Mexico and the Pecos River are provided by the International Boundary and Water Commission. Data at 14 sites in the Tennessee River Basin and another site in the Cumberland River Basin are provided by the Tennessee Valley Authority. The United States Army Corps of Engineers (USACE) shared data for nine sites they manage in the Cumberland River Basin. Recent records of daily discharge from the US Bureau of Reclamation supplement those from the USGS for sites on the Colorado and upper Rio Grande rivers. Potential errors associated with discharge measurements and implications to our results are discussed in the Supplementary Methods.

**Time series construction**. The overall study period spans 1 January 1920 to 31 December 2019 for which at least partial, extended (≥24 years) records of daily discharge are available at all sites. Time series of daily streamflow (in m$^3$ s$^{-1}$) are constructed based on data availability for each site of interest (Supplementary Data 2) and follows Déry et al.[47] in its approach. Daily discharge data sourced from the USGS, US Bureau of Reclamation, Tennessee Valley Authority, Nalcor Energy (Exploits River), USACE and NB Power are converted to metric units prior to analysis. For several waterways (e.g., the Nelson and Saguenay Rivers), data furthest downstream are first used, but when unavailable (prior to construction of dams and hydroelectric facilities), are replaced with those from the closest upstream gauging station while adjusting the data for the missing contributing area as necessary[47,48]. Gaps are in-filled with the mean daily discharge over the period of record; however, any calendar year with ≥10% missing records is excluded from analysis. Supplementary Data 2 lists the percentage of in-filled data at each site (average: 0.02%, maximum: 0.58%) omitting years when ≥10% of the data remain unavailable. Uncertainty in the results associated with data homogeneity and the gap-filling strategy is evaluated and discussed in the Supplementary Methods.

**Development of the WHI**. Various approaches are commonly used to explore flow alterations in regulated rivers including comparisons of hydrographs pre- and post-regulation[9,49,50], trends in peak and/or low flows[51] or of naturalized versus observed (regulated) flows[52–54]. A broader approach employs a set of multiple (up to 64) indicators of hydrologic alteration to quantify changes over the water year arising from regulation[55–57]. Another method combines hydrological data, reservoir information and a database of large dams in developing river regulation and fragmentation indices with a matrix of impact for application to all major global watersheds[4,5]. Apart from time domain analyses, Discrete Fourier Transforms or wavelet analyses offer additional insights on impacts of flow alterations from human interventions[15,22,46,58]. Consult Jumani et al.[40] for a review of river regulation and fragmentation indices including their applications, advantages and limitations.

While various approaches exist to infer hydrologic alterations from diversions, dam and reservoir operations including sub-daily hydropeaking cycles[59,60], none focuses on the weekly timescale, a primary periodicity of socioeconomic activity. Therefore, we develop a WHI that combines time and frequency domain terms to quantify weekly periodicity in river discharge. The time domain term ($T_T$, %) counts the number of weeks ($D_w$) in a given calendar year when two consecutive days exhibit flows lower than the corresponding weekly average ($\overline{Q_{1-7}}$), followed by five sequential days above the corresponding weekly average:

$$T_T = \max\left\{\frac{100}{52} \sum_{w=1}^{52} D_w,\ 0.001\right\} \text{ and where}$$

$$D_w = \begin{cases} 1 & \text{if } Q_{1,2} < \overline{Q_{1-7}} \text{ and if } Q_{3,\dots,7} > \overline{Q_{1-7}} \\ 0.25 & \text{if } Q_1 < \overline{Q_{1-7}} \text{ and if } Q_{2,\dots,7} > \overline{Q_{1-7}} \\ 0 & \text{if otherwise} \end{cases} \quad (1)$$

This sequence of daily flows is chosen to emphasize the typical weekly rhythm observed in hydropeaking rivers: low flows on weekends when electricity demand wanes, followed by high flows on weekdays when electricity demand waxes[9]. A partial score of 0.25 is ascribed to sites where six consecutive days above the weekly

average follow a single low flow day for that week. As some gauging sites lie downstream from points of regulation such that low flows are shifted later in the week rather than occurring on Saturdays and Sundays, we test all seven possible combinations of two consecutive days (e.g., Saturday/Sunday, Sunday/Monday, …, Friday/Saturday) and select the one that maximizes WHI at each site over the period of record. This approach for the time domain term attenuates the effects of cyclical (rather than periodic) variations from synoptic-scale storm activity, which otherwise leads to marked weekly cycles in pluvial rivers[46].

An application of Discrete Fourier Transforms to the daily discharge data provides the frequency domain term. Here we follow Wilks[61] in partitioning the daily discharge time series into sine and cosine waves of amplitude $C_k$ for harmonic $k$. Discrete Fourier Transforms are computed for each calendar year with the 52nd harmonic representing the weekly timescale of interest here. Then we compute the explained variance of the 52nd harmonic ($T_F$):

$$T_F = \frac{\left(\frac{n}{2}\right)C_{52}^2}{(n-1)s_Q^2} \quad (2)$$

where $n$ is the number of days in a given year (365 or 366 for a leap year), $C_{52}$ is the amplitude of the 52nd harmonic, and $s_Q$ is the standard deviation in discharge.

After expressing $T_T$ and $T_F$ as percentages, we take the base 10 logarithm of their product to obtain an annual WHI:

$$\text{WHI} = \log_{10}[B(T_T \times T_F)] \quad (3)$$

in which $B$ ($= 10$) is a coefficient chosen so that the median WHI $\approx 0$ among all 500 sites. Annual WHI values range typically from about $-4$ to $+4$ (although WHI values have no theoretical upper or lower bounds), with large positive values indicating strong weekly periodicity attributed to flow regulation at hydropower stations. In contrast, rivers with robust annual cycles with flows dominated by potent snowmelt-driven freshets and/or large (natural) storage capacity within abundant lakes, ponds and wetlands exhibit large negative WHI values. The transition between negative to positive WHI values marks a shift from annual to weekly dominant time scales of variability in flows. The 1980–2019 mean daily flows (considering the day of the week) for the Namakan River (Minnesota/ Ontario), St. Croix River (Maine/New Brunswick), and Smith River near Philpott (Virginia) illustrate the WHI ranging from the minimum, median, and maximum values (Supplementary Fig. 15). WHI values remain site-specific and must be interpreted with care, particularly moving away (both upstream and downstream) from measurement sites with an intervening body of water, a confluence or another point of regulation altering hydropeaking intensity.

**Statistical analyses**. We first compute WHI time series at all 500 sites (Supplementary Data 4) and develop a 'climatology' of index values for 1980–2019, with 14 years $\leq n_y \leq 40$ years depending on data availability at each site. Results for 1980–2019 are also tabulated in WHI bins of 0.75 across all sites, the USA and Canada. Summary statistics (mean, median, standard deviation, etc.) of the 1980–2019 WHI data are tabulated and their distribution tested for normality using the Shapiro-Wilk test. Similar climatological analyses are developed for each decade (1920s to 2010s) with results reported when $n_y \geq 5$ years at a given site. The Mann-Kendall test (MKT[62,63]) applied to all WHI time series with $n_y \geq 30$ years over 1980–2019 yields linear, monotonic trends in hydropeaking intensity, with $p < 0.05$ considered locally statistically significant. The field (or global) significance of the individual (or local) trend tests is assessed following Wilks[61]. The approach minimizes the false discovery rate (FDR) by first ranking $p$-values in ascending order for all trend tests with $n_y \geq 30$ years. Trends are then globally significant if $p < p_{FDR}$ depending on the distribution of sorted $p$ values as:

$$p_{FDR} = \max_{i=1,2,\dots,N}\{p_i : p_i \leq (i/N)\alpha_{\text{global}}\} \quad (4)$$

in which we set $\alpha_{\text{global}} = 0.10$. Trend analysis sensitivity to autocorrelation is tested in the Supplementary Methods. As the MKT does not distinguish between gradual versus abrupt changes in a variable, we implement the Pettitt test[21] while considering $p < 0.05$ as a break point in WHI time series. The year when the change point is identified along with the mean WHI prior to and after the break point years are tabulated.

We assess the 1920 to 2019 annual mean, median and maximum WHI across all sites with available data in a given year to track the overall evolution of hydropeaking intensity across the USA and Canada. We also count the annual number and percentage of sites that fall in the top decile of all 1920–2019 WHI scores. An additional metric reported is the discharge-weighted $\text{WHI}_{Qj}$ computed each calendar year (index $j$) as:

$$\text{WHI}_{Qj} = \sum_{i=1}^{n=500} \text{WHI}_{i,j} \times Q_{i,j} / \sum_{i=1}^{n=500} Q_{i,j} \quad (5)$$

where $Q_{i,j}$ (km$^3$ yr$^{-1}$) denotes the annual discharge and $i$ is the site index. This yields a relative measure of annual volumetric flows affected by weekly hydropeaking cycles rather than just the number of sites. For monotonic trend analysis, the MKT is applied to time series of overall mean annual WHI over the 1980–2019 focus period. The potential influence of missing data on the evolution of average WHI over 1980–2019 is assessed by substituting incomplete time series with each missing site's average WHI computed over the remainder of the focus

period. This yields an adjusted mean annual WHI time series for a first order assessment of the influence of incomplete data.

A histogram illustrates the distribution of two consecutive days when low flows emerge relative to the expected value of $1/7 = 0.143$ were these randomly distributed. Fractions of the seven possible two-day combinations are partitioned according to WHI $\gtrless 0$. The histogram also includes the corresponding mean WHI across all rivers for a given two-day combination of low flows. A Chi-Square goodness-of-fit test[64] verifies the hypothesis of whether the distribution of low flow days differs significantly from the expected value with threshold $p = 0.05$. Similarly, we test if the corresponding mean WHI values for the two-day pairs with low flows follow a uniform distribution using a Chi-Square test. The relationship between annual WHI values and mean annual flows over 1980–2019 is evaluated using Pearson's correlation coefficient with $p < 0.05$ considered statistically significant values. Next, we transform annual discharge time series to standardized anomalies over the period of record at each site (with <10% missing data in a calendar year). Decadal mean standardized anomalies for all available sites are then computed when $n_y \geq 5$ years in a given decade. These decadal average anomalies are binned in increments of 0.25 standard anomaly for comparison with WHI decadal distributions.

The influence of dams on the temporal evolution of WHI values is assessed using 14 hydroelectric facilities managed by the Tennessee Valley Authority (TVA;[65] Supplementary Data 3). Here, we take the year a project was completed as its commissioning year to establish the response of the WHI to flow regulation. Then, we report the influence of 14 multi-purpose reservoirs[66] including those managed for hydropower production on the WHI computed for sites on downstream waterways.

To explore possible factors contributing to WHI trends we assess whether the dispersion of flows across the seven days of the week is changing over time. Here, we first compile total annual flows (in m$^3$ s$^{-1}$) for each of the seven days of the week, as well as the overall average, over each calendar year. Then, we quantify departures (as a percentage) for each day of the week relative to the annual mean. Next, we calculate standard deviations ($\sigma$) in the percentage departures for the seven days of the week each year, creating $\sigma$ time series for all 500 sites over 1980–2019. Finally, application of the MKT on the $\sigma$ time series (when $n_y \geq 30$ years) yields 1980–2019 dispersion trends.

## Data availability

Data related to this article can be found in the Supplementary Data files. Discharge data used in this study are available in the following publicly accessible databases: Centre d'Expertise Hydrique du Québec (http://www.cehq.gouv.qc.ca/hydrometrie/ historique_donnees/info_validite.htm), US Bureau of Reclamation (https://data.usbr.gov/), United States Geological Survey (https://waterdata.usgs.gov/nwis), Water Survey of Canada's Hydrometric Database (https://wateroffice.ec.gc.ca), and the International Boundary and Water Commission (https://www.ibwc.gov/Water_Data/). For some regulated rivers, proprietary or unpublished discharge data can be requested from the following data providers: BC Hydro, Evolugen, H2O Power, Hydro-Québec, International Boundary and Water Commission, Manitoba Hydro, Nalcor Energy, NB Power, Ontario Power Generation, Rio Tinto, Tennessee Valley Authority, TransAlta, and USACE (see Supplementary Data 3). Source data are provided with this paper.

## Code availability

The Fortran code used in this study is available online with an explanation at https:// doi.org/10.5281/zenodo.5646458.

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

## Acknowledgements

Thanks to the Water Survey of Canada and its provincial and territorial partners, the Centre d'Expertise Hydrique du Québec, USGS, BC Hydro, Evolugen by Brookfield Renewable, TransAlta, Manitoba Hydro, Ontario Power Generation, $H_2O$ Power, Rio Tinto, Hydro-Québec, NB Power, Nalcor Energy, the Tennessee Valley Authority, the International Boundary and Water Commission, USACE and the US Bureau of Reclamation for providing hydrometric data. Thanks to Aseem Sharma (UNBC/NRCan) for preparing the spatial plots, Clyde McLean and Joanna Barnard (Nalcor Energy), Jim Samms (NB Power), Marie Broesky, Kevin Gawne, Kristina Koenig, Phil Slota, Kevin Sydor, Efrem Teklemariam, Mike Vieira and Shane Wruth (Manitoba Hydro), Matt MacDonald (Ontario Power Generation), Erik Richards and Marc Mantha (H2O Power), Samer Alghabra and Mokhtar Moujahid (Hydro-Québec), Bruno Larouche and Richard Loubier (Rio Tinto), Michael Smilski (TransAlta), Jim Li, Debbie Rinvold and Stephanie Smith (BC Hydro), Adrian Cortez and Delbert Humberson (International Boundary and Water Commission), Kelly Withers and Matti Hanninen (Evolugen), and Robert Dillingham (USACE) for providing comments on this work and for additional data for regulated rivers, Dwayne Akerman, Amber Brown, Michel Desjardins, Matt Falcone, Samantha Hussey, Lyssa Maurer, Angus Pippy, Melanie Taylor, and Frank Weber (Water Survey of Canada) for sharing supplemental hydrometric data, and Huilin Gao (Texas A&M), John Zhu (Texas Water Development Board), Rajtantra Lilhare (UNBC), Julie Thériault (UQAM) and Mike Vieira and Kristina Koenig (Manitoba Hydro) for logistical support. This research was supported by the Natural Sciences and Engineering Research Council of Canada, Manitoba Hydro, and partners through funding of the BaySys project.

## Author contributions

S.J.D. designed the study, extracted hydrometric data and constructed time series of daily discharge for all rivers, formulated the weekly hydropeaking index, developed the codes, performed the statistical and computational analyses, and drafted line graphs with support from M.A.H.H., T.A.S., and T.J.T. S.J.D. wrote the manuscript with contributions from all co-authors and all contributed to manuscript refinement and revisions.

## Competing interests

The authors declare no competing interests.
