## [Peer Review File · Nature Communications]

Vanishing weekly hydropeaking cycles in American and Canadian riversReviewers' Comments:

Reviewer #1:

Remarks to the Author:

Authors analyze daily observed flow across 100 years over an increasing number of river sites, up to 400 in 2019. They specifically look into the weekly Hydropeaking Index, a novel index to quantify the week-end vs weekday river alterations. The robust statistical analysis demonstrates that the alterations increased from 1920 to about 1990, plateaued for a bit and have decreased since the 2010s. Authors discuss potential reasons for this decrease in the last decade, which include changes in demand, environmental regulation and new generation resources.

Organization

- The results and discussion section need re-organization. The results section presently focus on plain statistics with no maps, and lead to so many technical questions that are only answered in the SI and with no actually insight of what the actual result to be promoted is.

- Some section in the methods and SI would actually enhance the flow of the paper. For example, the description of the area is necessary to support the current description in the results section. Also Fig 2 in SI actually tells the story of the paper and is more impactful than some of the maps in the main manuscript that tend to only show the data and support some specific examples.

Technical Approach and impact of the paper on the community

- In the introduction (L81-84), authors "conclude" that the "vanishing " is due to changes in hydropower demand, environmental regulation and new generation resources. However those causes were not fully demonstrated, only discussed. I would suggest the authors to reframe this sentence with "probably due to .." but most importantly focus on the impact of why it matters. May I suggest that this matters for the hydropower industry long term planning, but also for the power system operators. Specifically "Does that mean that hydropower is "less flexible" or does that mean that hydropower flexibility is used differently?" The potential reasons brought forward by the authors could be categorized based on "who is affected by that result" , or other ways, to provide more clarity on why this paper is important.

- The analysis focused on 400 sites, and authors discussed governances. In order to enhance the impact of the paper (who should be concerned by this result), showing trends in WHI by main river basin (hydrology, environmental regulation, level of regulation, etc) and by market regions, or grid, would provide more support to the discussion of potential causes for the regional trends. It would be more informative than by latitude and longitudes.

- "Hydropower demand" throughout the paper - it would be more accurate to say "electricity demand " that is changing due to changes in socio-economic development etc. The hydropower contribution (or generation) however is indeed influenced by the changes in generation portfolio, markets, environmental regulation and so on.

- More potential causes- for example it is possible that with wind and solar the prices differences have changed and hydropower provide new types of services, such as capacity markets, which could affect the WHI index. Socio-economic development is pretty vague and could mean changes in water demands in general?

- Authors presently mention that "spilling" is the reason for lower WHI during wet years. During a wet year, especially snowmelt period, the hydropower operators generate firm energy, i.e. reduced sub-daily peaking and very limited to none week-end/week day alterations. - it should be revised for completeness in the manuscript.

Editing

- In concluding remarks, L399, it

- L429 - specify discharge at a daily time scale.

- L512-514 - this is a nice and succinct description that could have made its way in the main part of the manuscript along with the description of the domain.

- L524: DTF - spell out

- L571 - WHI_q was mentioned in the results section with no description. Again, figure 2 of SI would

help in describing (and synthetizing) the impactful-take home message results.

Reviewer #2:

Remarks to the Author:

The manuscript developed a novel weekly hydropeaking index for quantifying the 1920-2019 intensity and prevalence of hydropeaking cycles at 400 sites across the United States of America and Canada. The key finding is that there is a recent decline in weekly hydropeaking cycles in the US and Canada. More importantly, the findings may have a broad impact across multiple disciplines. On one hand, the causes of this declined weekly hydropeaking cycles can be attributed to factors from changing climate, socioeconomic shifts, alternative energy production, to legislative and policy changes. On the other hand, it has very significant ecohydrological implications. In short, the manuscript has revealed an important area which has a lot of potential to be explored in many ways. The manuscript is overall well-organized and well-written. The new index can be easily adopted in other regions across scales as long as long-term daily streamflows observations are available.

There are a few areas which can be improved.

- 1) It would be nice to compare the WHI before and after reservoir constructions. Since the weekly hydropeaking cycles are directly driven by reservoir flow regulations, the first thing to check how it has changed after dam construction. Reservoir info can be acquired from databases such as GRanD.
- 2) While most reservoirs have multiple functions, the manuscript has attributed the changes of hydropeaking cycles to hydropower generation. Therefore, additional analysis which compares the WHI downstream of different types of reservoirs (by primary function) would be interesting. For instance, how do the WHI values downstream of irrigation reservoirs compare to those downstream of hydropower reservoirs?
- 3) Some more quantitative investigations about the causes of the changed WHI would be necessary. Currently, multiple drivers for the decline have been pointed out. However, there is a lack of evidence on this regard. For instance, it is unclear what time period, spatial domain does the "above average precipitation" refer to. Although the alternative energy has increased, the hydropower generation hasn't decreased much. In this sense, the flow regulation may not have changed much. Then, how to relate alternative energy to the finding?

From Huilin Gao

Reviewer #3:

Remarks to the Author:

place this table by a figure.

Review of paper “Vanishing weekly hydropeaking cycles in American and Canadian rivers”

The authors propose a new WHI index to analyze weekly fluctuations in daily flows in regulated rivers (400 sites) in the United States and Canada to illustrate the decrease in flows that take place on weekends (Saturday and Sunday) downstream from dams. Results from analyzing flows at 400 sites over the 1920-2019 period show that there is an increase in the number of sites showing such decrease in flows since 1920, reaching a maximum in 1963, followed by a significant decline until 2019. These changes are due to climate- and human-related factors.

Originality of the work

The development of the WHI index and its application to a large number of sites in the United States and Canada in order to highlight this decrease in stream flows is, in my mind, a perfectly original scientific contribution to the study of the impacts of dams worldwide. In addition, the issue of flow fluctuations on weekend days is also an original contribution to the study of the impacts of dams. The authors have shown that this variation in flows results from the interaction of numerous climate and human factors, thereby highlighting the complex nature of factors affecting streamflow downstream from dams. There is no doubt that the results are of great scientific interest for understanding better the impacts of flow management on the function and hydromorphological and hydroecological evolution of stream ecosystems downstream from dams. As such, they are contributing to the development of flow requirements for the management, restoration and conservation of these anthropized ecosystems.

Review of the paper

1. Statistical methods used

- Regarding the interannual variability of the WHI index, the authors only considered the influence of autocorrelation on the significance of the MK (Mann-Kendall) test even though, when analyzing long data series (1920-2019), they definitely should consider the influence of the persistence phenomenon affecting the long series by applying the LMK test. In fact, the presence of persistence weakens the MK test (see, among others, Kumar et al., 2009; Dinpashoh et al., 2014).
- The MK method does not detect breaks in mean values nor discriminate between sharp and gradual variance values. A statistical test that allows objective detection of breaks in mean values of the analyzed series (e.g., Pettitt, Lombard test) must be used, making it possible to determine rigorously the dates of shifts in mean values of the WHI index.
- It would be useful to apply a classification method (e.g., bottom-up hierarchical classification) to WHI index values to subdivide the 400 sites in classes to improve their characterisation and description.

2. Results

Table 1 should be replaced with a table showing the result of all values of the WHI index derived for the 400 sites. See an example

Indice	Number of sites	% of sites	Sites in USA (%)	Sites in Canada (%)
>3.5				
3.0 – 3.5				
2,5 – 3.0				
2.0 – 2.5				
1.5 – 2.0				
1.0 1.5				
0.5 – 1.0				
0.0 – 0.5				
-0.5 – 0.0				
-1.0 - -0.5				
-1.5 - -1.0				
-2.0 - -1.5				
-2.5 - -2.0				
-3.0 - -2.5				
-3.5 - -3.0				
< -3.5				

N.B. It may be preferable to replace this table by a figure.

REVIEWER COMMENTS

Reviewer #1 (Remarks to the Author):

Authors analyze daily observed flow across 100 years over an increasing number of river sites, up to 400 in 2019. They specifically look into the weekly Hydropeaking Index, a novel index to quantify the week-end vs weekday river alterations. The robust statistical analysis demonstrates that the alterations increased from 1920 to about 1990, plateaued for a bit and have decreased since the 2010s. Authors discuss potential reasons for this decrease in the last decade, which include changes in demand, environmental regulation and new generation resources.

We sincerely thank Reviewer #1 for the thoughtful and constructive comments provided in this report on our paper. We address each of Reviewer #1's comments with a point-by-point response in bold lettering. Thank you for taking the time to carefully read and review our manuscript as we feel that by addressing these comments our paper is now much stronger.

Organization

- The results and discussion section need re-organization. The results section presently focus on plain statistics with no maps, and lead to so many technical questions that are only answered in the SI and with no actually insight of what the actual result to be promoted is.

We acknowledge that the prior version of the manuscript included a considerable amount of statistics on the WHI. Given the WHI is a novel metric introduced in this paper, we judge it critical to report on the overall statistics of the WHI so that the reader can gauge its central tendency, dispersion, and range across the 500 sites of interest and over time. Furthermore, these statistics are first reported in the Results section (now after the study area description) to provide baseline information for the interpretation of the spatio-temporal variability of the WHI across the USA and Canada. Thus the revised text retains a subset of the statistics reported in an earlier version of the paper; however, we have shifted the result of the Shapiro-Wilk test to the Supplementary Information. As well, we have deleted the Cullen and Frey graph (Supplementary Figure 1 in the initial submission), and the skewness and kurtosis values to reduce the amount of statistics presented in the first part of the Results section. Other parts of the paper have also been revised and restructured to further promote the key message of declining WHI trends across the USA and Canada.

Three of the four main figures in the original paper are maps depicting spatial plots of the WHI. Thus there are quite a few maps (along with several others in the Supplementary Information) to visualize and interpret the WHI results aside from the statistics presented in the paper. Due to the strict length limitation of *Nature Communications* (main text of 5,000 words or less) and our strong desire to provide a comprehensive analysis of the WHI results, many technical aspects are relegated to the Supplementary Information. Our Results section builds strongly the case for the study's main findings of vanishing weekly hydropeaking cycles in the USA and Canada, namely in the temporal evolution and trend

analysis subsection. Our findings are further promoted in the Discussion where we invoke several possible reasons for the declines in weekly hydropeaking cycles in the study domain.

Please also see our response to the next comment regarding some additional restructuring of the paper.

- Some section in the methods and SI would actually enhance the flow of the paper. For example, the description of the area is necessary to support the current description in the results section. Also Fig 2 in SI actually tells the story of the paper and is more impactful than some of the maps in the main manuscript that tend to only show the data and support some specific examples.

We agree some of the material in the Supporting Information (previously Supplementary Figure 2) and in the Methods is more appropriate for the main body of the paper. In response to this comment, we have now shifted content describing the study area previously introduced in the Methods to the first paragraph in the Results section (lines 93-105). The plot illustrating the time series of mean annual WHI and other important metrics now forms part of the Results section (as the revised Figure 3) rather than in the Supplementary Information.

Technical Approach and impact of the paper on the community

- In the introduction (L81-84), authors “conclude” that the “vanishing” is due to changes in hydropower demand, environmental regulation and new generation resources. However those causes were not fully demonstrated, only discussed. I would suggest the authors to reframe this sentence with “probably due to ..” but most importantly focus on the impact of why it matters. May I suggest that this matters for the hydropower industry long term planning, but also for the power system operators. Specifically “Does that mean that hydropower is “less flexible” or does that mean that hydropower flexibility is used differently?” The potential reasons brought forward by the authors could be categorized based on “who is affected by that result” , or other ways, to provide more clarity on why this paper is important.

Indeed, our work only discusses possible reasons leading to recent declines in weekly hydropeaking cycles across Canada and the USA. Without an extensive database of electricity generation and other information regarding socioeconomic activity, policy and governance changes (among other factors), it is impossible to pinpoint the exact causes of the recent WHI declines reported in our work. Nonetheless, we anticipate this will motivate other studies that may well tackle these issues in a more comprehensive way, as this remains beyond the scope of the present effort. Note, however, that in response to a comment from Reviewer #2 we have inserted some additional material to back up statements in regards to possible factors leading to recent WHI declines.

To address this comment, we have replaced the word “conclude” with “propose” on line 86 and inserted “likely” before “contributing factors” on line 89 in the final paragraph of the Introduction.

As raised by Reviewer #1, the previous version of the manuscript did not properly convey why this work is important in the Introduction. In response to this, we have added a sentence at the end of the third paragraph of the Introduction that reads as follows: “Aside from their ecohydrological impacts, changes in hydropeaking remain key concerns for long term planning within the hydropower industry, system operators, and water resources managers.”

- The analysis focused on 400 sites, and authors discussed governances. In order to enhance the impact of the paper (who should be concerned by this result), showing trends in WHI by main river basin (hydrology, environmental regulation, level of regulation, etc) and by market regions, or grid, would provide more support to the discussion of potential causes for the regional trends. It would be more informative than by latitude and longitudes.

Water management and governance occur on multiple levels leading to complex interactions with impacts to hydropower production and hence hydropeaking cycles. Indeed, water management and governance occur within and across legislative / political boundaries (e.g. provincial, state-wide, federal), at the watershed scale, and by market regions. Given this work emphasizes hydropeaking cycles emerging from hydropower production, we have added to the map of the study area the principal power grid interconnections of the USA and Canada (see Supplementary Figure 14a).

To avoid cluttering the plots depicting spatial results for the WHI values, only the map of the study area shows the power grid interconnections. However, the spatial plots retain the primary provincial, state, and federal boundaries to allow interpretation of the results across political jurisdictions. Primary waterways are also identified on the maps to infer results at the watershed scale. The paper retains Supplementary Figure 14b-e illustrating the distribution of latitudes, longitudes, gauged areas and mean annual discharges for the 500 sites as primary metadata to our analyses.

Cross comparison of the primary power grid interconnections (Supplementary Figure 14a) with the map depicting spatial trends in WHI (Figure 5) reveals that most of the negative WHI trends lie within the Western Electricity Coordinating Council (WECC), Northeast Power Coordinating Council (NPCC) and SERC Reliability Corporation (SERC) synchronous grids. Therefore, a statement has been added at lines 232-234 that identifies the primary power grid interconnections where negative WHI trends emerge: “Clusters of negative WHI trends lie primarily within the Western Electricity Coordinating Council, Northeast Power Coordinating Council and SERC Reliability Corporation power grid interconnections.” In a future effort, we anticipate tackling in more detail the causality of recent declines in weekly hydropeaking cycles including links to market regions and power grid interconnections.

- “Hydropower demand” throughout the paper - it would be more accurate to say “electricity demand” that is changing due to changes in socio-economic development etc. The hydropower contribution (or generation) however is indeed influenced by the changes in generation portfolio, markets, environmental regulation and so on.

We agree with the reviewer’s comment that it is not only hydropower demand that is changing in response to socioeconomic development and other factors. As such we have replaced the word “hydropower” with “electricity” where relevant to generalize this statement throughout the manuscript.

- More potential causes– for example it is possible that with wind and solar the prices differences have changed and hydropower provide new types of services, such as capacity markets, which could affect the WHI index. Socio-economic development is pretty vague and could mean changes in water demands in general?

Recent reductions in the costs for solar and wind energy production may have resulted in hydropower providing new types of services such as capacity markets. A statement to that effect is now included in the first paragraph of the discussion (see lines 312-314). The term “socioeconomic development” is purposefully vague to account for the multitude of factors (e.g. a shifting manufacturing sector, globalization, lifestyle changes, commercial and industrial activity, etc.). Rather than attempting to provide an exhaustive list of these potential factors, the text retains the term “socioeconomic factors” in several instances.

- Authors presently mention that “spilling” is the reason for lower WHI during wet years. During a wet year, especially snowmelt period, the hydropower operators generate firm energy, i.e. reduced sub-daily peaking and very limited to none week-end/week day alterations. – it should be revised for completeness in the manuscript.

This is a very interesting point raised by Reviewer #1 that certainly merits further investigation. Some preliminary analyses do suggest that the weekly hydropeaking cycles experience seasonality, perhaps as hydropower generating stations shift from peaking plants to firm energy production depending on water availability. We anticipate tackling this issue in a future effort for which we will explore the seasonality of the WHI across all sites depending on water availability. In the meantime, to ensure completeness of the manuscript in regards to the impacts of wet/dry spells on hydropeaking cycles, a statement has been added to the Discussion (lines 333-335) as follows: “Alternatively, wet years may lead utilities to generate continuous baseload energy instead of peaking hydropower, inducing a similar effect.”

Editing

- In concluding remarks, L399, it

It is unclear from this statement what text should be modified on line 399 of the manuscript. As such, no modification to the text is implemented based on this comment.

- L429 – specify discharge at a daily time scale.

We have inserted “daily” prior to “discharge” on line 467.

- L512-514 – this is a nice and succinct description that could have made its way in the main part of the manuscript along with the description of the domain.

Thank you for this comment, we have incorporated a similar statement in the third paragraph of the Introduction (see lines 80-82): “We show that the WHI captures well the typical weekly rhythm observed in hydropeaking rivers, with low flows on weekends when hydropower demand wanes then high flows on weekdays when hydropower demand waxes.”

- L524: DTF – spell out

The abbreviation “DFT” for Discrete Fourier Transform has been deleted from the paper including at lines 535, 563, and 565-566.

- L571 – WHI_q was mentioned in the results section with no description. Again, figure 2 of SI would help in describing (and synthesizing) the impactful-take home message results.

Thank you for this remark. A definition of the discharge-weighted WHI is now included where this is introduced in the Results section (see lines 174-177). For completeness, the mathematical definition (Equation 5) is retained in the Methods section of the paper (line 617). The plot depicting the time series of various WHI metrics including the discharge-weighted values is now part of the main paper (as revised Figure 3).

Reviewer #2 (Remarks to the Author):

The manuscript developed a novel weekly hydropeaking index for quantifying the 1920-2019 intensity and prevalence of hydropeaking cycles at 400 sites across the United States of America and Canada. The key finding is that there is a recent decline in weekly hydropeaking cycles in the US and Canada. More importantly, the findings may have a broad impact across multiple disciplines. On one hand, the causes of this declined weekly hydropeaking cycles can be attributed to factors from changing climate, socioeconomic shifts, alternative energy production, to legislative and policy changes. On the other hand, it has very significant ecohydrological implications. In short, the manuscript has revealed an important area which has a lot of potential to be explored in many ways. The manuscript is overall well-organized and well-written. The new index can be easily adopted in other regions across scales as long as long-term daily streamflows observations are available.

Our sincere thanks to Reviewer #2 for this positive overview of our manuscript and for the most helpful and constructive comments provided in this report. We address each of the three comments with point-by-point responses below using bold text. Given in part the request to include some additional results on the impacts of dams and reservoirs on the weekly hydropeaking index (WHI) values, our database has been augmented to 500 sites. By addressing these comments, we firmly believe our paper has significantly improved. Thank you kindly for your time and effort in providing this report on our manuscript.

There are a few areas which can be improved.

1) It would be nice to compare the WHI before and after reservoir constructions. Since the weekly hydropeaking cycles are directly driven by reservoir flow regulations, the first thing to check how it has changed after dam construction. Reservoir info can be acquired from databases such as GRanD.

As we compute the weekly hydropeaking index (WHI) each year at all sites, it responds to the commissioning of new hydropower infrastructure and other factors that impact hydropeaking. This is evident in Supplementary Figure 13 that illustrates how rivers of northern Canada including the Churchill, La Grande, Nelson and Peace, jump from large negative to large positive WHI values once hydropower dams are commissioned. Vertical red lines denoting the years when hydropower facilities were commissioned have been added to this plot to better illustrate this feature in our WHI time series.

Given the importance of hydropower infrastructure on the WHI results, we have added a new paragraph in the Results section addressing the influence of dam commissioning on WHI evolution. Specifically, we now present time series of annual WHI for 1920-2019 at 14 hydroelectric dams operated by the Tennessee Valley Authority (TVA). Most of these dams were built in the early part of the 20th century and upon commissioning yield abrupt inclines in WHI (Supplementary Figure 5 reproduced as Figure R1 below). For instance, the inception of the Blue Ridge Dam on the Toccoa (Ocoee) River induces a sharp rise in WHI from 0.464 in 1930 to 2.684 in 1931, with elevated WHI scores thereafter. At other sites such as Apalachia Dam on the Hiwassee River, however, dam commissioning leads to

little change in WHI. Apalachia Dam is a run-of-river facility with flows regulated mainly at the upstream Hiwassee Dam that has considerable flood storage capacity (0.253 km³). Lines 243-252 in the Results section describe how dam commissioning and operation dictate WHI evolution in regulated waterways in the Tennessee Valley.

Figure R1: Temporal evolution of annual WHI at 14 sites with hydropower dams managed by the Tennessee Valley Authority, 1920-2019. Vertical red lines denote commissioning years of hydropower dams at the gauging site (<https://www.tva.com/energy/our-power-system/hydroelectric>). Note that Great Falls Dam on the Caney Fork River was commissioned in 1916 explaining the absence of a vertical red line in that panel.

2) While most reservoirs have multiple functions, the manuscript has attributed the changes of hydropeaking cycles to hydropower generation. Therefore, additional analysis which compares the WHI downstream of different types of reservoirs (by primary function) would be interesting. For instance, how do the WHI values downstream of irrigation reservoirs compare to those downstream of hydropower reservoirs?

The presence of an upstream reservoir from a gauging site may also influence the weekly hydropeaking cycles. Here, we employ a subset of 14 waterways with different types of reservoirs examined by Ferrazzi et al. (2021). For sites with upstream reservoirs managed, at least in part, for hydropower production, the WHI generally stays elevated at positive values. In contrast, for sites downstream of reservoirs serving other functions (see Table

R1), WHI oscillates near zero (see Figure R2). Thus the type of reservoir along with dam operations play a distinct role on the temporal evolution of the WHI.

Table R1: Alphabetical list of 14 reservoirs with gauging sites on rivers part of our extended database (information sourced from Ferrazzi et al. 2021).

Reservoir	River	Capacity (Mm ³)	Type*
Allegheny	Allegheny	1,460	FPAQRW
Cannonsville	WB Delaware	362	S
Carters	Coosawattee	583	FP
Cave Run	Licking	757	FQRW
Green	Green	892	FSAQR
Mark Twain	Salt	1,760	FNPRSW
Perry	Delaware	950	FSRWX
Philpott	Smith	393	FPR
Pomme de Terre	Pomme de Terre	802	FRWX
Raystown	Juniata	940	FPRW
Shelbyville	Kaskaskia	844	FSNRW
Stockton	Sac	2,060	FPRW
Waterbury	Little	46	FRP
Zoar	Housatonic	33	P

*Reservoir functions are: flood control (F), urban water supply (S), hydropower production (P), low flow augmentation (A), navigation (N), wildlife preservation (W), water conservation and sedimentation (X), water quality control (Q), and public recreation (R).

Figure R2: Temporal evolution of annual WHI at 14 sites with upstream reservoirs with different functions (see Table R1), 1920-2019. Red lines denote sites with an upstream reservoir managed, at least in part, for hydroelectricity production.

We have added text on the impacts of reservoirs, depending on their functions, to the Results section following the description of dam commissioning and operation, on lines 252-256. Table R1 and Figure R2 are also included in the Supplementary Information (Supplementary Table 7 and Supplementary Figure 6).

3) Some more quantitative investigations about the causes of the changed WHI would be necessary. Currently, multiple drivers for the decline have been pointed out. However, there is a lack of evidence on this regard. For instance, it is unclear what time period, spatial domain does the “above average precipitation” refer to. Although the alternative energy has increased, the hydropower generation hasn’t decreased much. In this sense, the flow regulation may not have changed much. Then, how to relate alternative energy to the finding?

Correct, we invoke in our Discussion a number of potential causes for the recent WHI declines. This includes: 1) a generally wet decade in the 2010s; 2) socio-economic shifts such as increases in commercial and industrial activity on weekends; 3) shifts towards other modes of electricity production (e.g. renewable resources such as wind, wave and solar energy); 4) deregulation of electricity production and power grid interconnections; 5) legislative and policy changes affecting water management (e.g. increased concern for

ecological, environmental and cultural flows). We concur that providing more concrete evidence for the reductions in WHI is highly desirable; however, obtaining all of the data to undertake these analyses is beyond the scope of the present effort.

To address the Reviewer's comment, we provide some additional information in regards to points 1), 3), and 4) outlined in the previous paragraph. First, we have added the spatial distribution of the standardized discharge anomalies in the 2010s, which was a wet decade relative to others between the 1920s to 2000s across the northern two-thirds of the study area (see Supplementary Figure 8b included as Figure R3 below). As discussed in the paper, this may have led dam operators to spill greater amounts of water and/or to generate firm energy rather than peaking hydropower, abating the weekly hydropeaking cycles.

Figure R3: Spatial distribution of decadal standardized discharge anomalies at 500 sites across the USA and Canada, 2010-2019. Negative values indicate relatively dry conditions while positive values denote relatively wet conditions.

Furthermore, there has been explosive growth in the generation of electricity from non-hydro renewable resources in the last decade in both the USA and Canada (Figure R4). Indeed, non-hydro renewable energy production jumped from 203.5×10^6 kWh in 2010 to 603.6×10^6 kWh in 2020 by which time it comprised 12.9% of overall electricity generation in the USA and Canada combined. The emerging availability of renewable sources of electricity such as solar, wind and wave energy diminishes the reliance on hydropower to match peak demand. For instance, solar energy potential peaks during midday when electricity demand is high. It is also evident in Figure R4 that the production of hydroelectricity has remained stable in the USA but continued to expand in Canada through the 2010s. Indeed, the generation of hydropower increased by 66.0×10^6 kWh between 2010 and 2020 in the USA and Canada combined even as the weekly hydropeaking cycles diminished in intensity.

Figure R4: Annual cumulative electricity generation (kWh, left or %, right) for four types of electricity production in (a, b) the USA, (c, d) Canada, and (e, f) the USA and Canada combined, 1980-2020. Note the different y-axis scales in panels (a), (c) and (e). There is a rapid expansion of non-hydro renewable sources of electricity in the 2010s across all regions. Data are sourced from the U.S. Energy Information Administration (<http://iea.org>).

Plotting the annual electricity production from non-hydro renewable sources vs. the mean annual WHI across Canada and the USA over a 40-year period reveals a statistically-significant anti-correlation (Figure R5). While this statistical relationship does not equate to cause and effect, it does suggest that the rapid emergence of non-hydro renewable sources of energy may play a leading role in the diminishing weekly hydropeaking cycles.

Figure R5: Annual non-hydro electricity production across the USA and Canada combined vs. the mean annual WHI at 500 sites, 1980-2019. The thick line denotes the linear regression with $R = -0.82$, $p < 0.05$.

Synchronous power grid interconnections, deregulation and the centralization of electricity dispatching may also yield reductions in WHI. As reported in the Discussion, the commissioning of the Churchill Falls hydropower plant in the early 1970s followed by the James Bay Hydroelectric Complex in the early to mid-1980s shifted the presence of hydropeaking from rivers in southern to northern Québec and Labrador. With the continued expansion of electricity exports from Canada to the USA (see Figure R6) through the international power grid interconnections, this also likely precipitated a reduction in the number of hydropeaking sites in the northern USA.

Figure R6: Total annual electricity exports (10⁶ kWh) from Canada to the USA, 1980-2019. Data are sourced from the U.S. Energy Information Administration (<http://iea.org>).

Based on these examples, there is evidence that the relatively wet 2010s, the expansion of non-hydro renewable sources of electricity, and integration within the electricity markets are contributing factors to diminishing weekly hydropeaking cycles in parts of North America. Socioeconomic determinants along with legislative, policy and water management changes do require additional investigation to confirm their role in vanishing weekly hydropeaking cycles. Nevertheless, our study provides two concrete examples on how modifications in water management influences the WHI: 1) the Sturgeon River in northern Michigan's Upper Peninsula experienced a sharp decrease in WHI when the Prickett hydroelectric facility switched operations from peaking to run-of-river power production

(Auer et al. 1996); 2) the Churchill River at Churchill Falls Powerhouse in Labrador observed a step decrease in WHI in 1997 related to electricity markets and a change in water management in that system (see Supplementary Figures 10 and 13).

The revised manuscript now includes these additional sources of information to further back the statements on possible causes for the vanishing weekly hydropeaking cycles in the USA and Canada.

From Huilin Gao

We sincerely thank Dr. Gao for these constructive comments that has led to a much improved paper.

Reviewer #3 (Remarks to the Author):

Review of paper “Vanishing weekly hydropeaking cycles in American and Canadian rivers”

The authors propose a new WHI index to analyze weekly fluctuations in daily flows in regulated rivers (400 sites) in the United States and Canada to illustrate the decrease in flows that take place on weekends (Saturday and Sunday) downstream from dams. Results from analyzing flows at 400 sites over the 1920-2019 period show that there is an increase in the number of sites showing such decrease in flows since 1920, reaching a maximum in 1963, followed by a significant decline until 2019. These changes are due to climate- and human-related factors.

Originality of the work

The development of the WHI index and its application to a large number of sites in the United States and Canada in order to highlight this decrease in stream flows is, in my mind, a perfectly original scientific contribution to the study of the impacts of dams worldwide. In addition, the issue of flow fluctuations on weekend days is also an original contribution to the study of the impacts of dams. The authors have shown that this variation in flows results from the interaction of numerous climate and human factors, thereby highlighting the complex nature of factors affecting streamflow downstream from dams. There is no doubt that the results are of great scientific interest for understanding better the impacts of flow management on the function and hydromorphological and hydroecological evolution of stream ecosystems downstream from dams. As such, they are contributing to the development of flow requirements for the management, restoration and conservation of these anthropized ecosystems.

Thank you for this positive overview of our manuscript and for highlighting the novel aspects of this research. We address the comments in this report point-by-point below using a bold font. We appreciate the very informative and constructive comments provided by Reviewer #3 on our work, which has led to a much improved paper.

Review of the paper

1. Statistical methods used

- Regarding the interannual variability of the WHI index, the authors only considered the influence of autocorrelation on the significance of the MK (Mann-Kendall) test even though, when analyzing long data series (1920-2019), they definitely should consider the influence of the persistence phenomenon affecting the long series by applying the LMK test. In fact, the presence of persistence weakens the MK test (see, among others, Kumar et al., 2009; Dinpashoh et al., 2014).

Thank you for this comment on the influence of autocorrelation and long-term persistence on trend detection using the Mann-Kendall test (MKT). Please first note that the MKT is applied only to the focused study period of 1980-2019 and not to the entire century for which at least partial data are available at the selected sites (see Figure 5). Nonetheless, we concur that serial correlation can diminish the true significance of the monotonic trends inferred by MKT. To that end, we follow Yue et al. (2002) in removing the lag-1

autocorrelation, or AR(1), in the WHI time series when statistically-significant ($p < 0.05$) local trends are inferred.

Following the steps outlined by Yue et al. (2002), we obtain 26 sites where the detrended WHI times series have statistically-significant AR(1) values in the presence of locally statistically-significant MKT trends. After pre-whitening the time series we find that only one site, the English River at Manitou Falls, no longer exhibits a locally statistically-significant trend ($p = 0.065$). Thus mitigating the effects of the lag-1 serial correlation on the trend analysis does not alter our main conclusion that a large number of sites across the USA and Canada exhibit significant declines in WHI from 1980-2019.

As an example, Figure R7 illustrates the original WHI time series at four sites exhibiting statistically-significant lag-1 autocorrelations as well as positive (b and c) and negative (a and d) trends ($p < 0.05$) based on the MKT. The trend-free pre-whitening of the WHI time series according to the methodology outlined by Yue et al. (2002) removes the lag-1 serial correlation component while retaining the linear trends.

Figure R7: Time series of the original and pre-whitened WHI for (a) the Chattahoochee River, (b) Colorado River at Lees Ferry, (c) English River at Manitou Falls, and (d) Kootenai River, 1980-2019. While all original, detrended WHI time series exhibit statistically-significant AR(1), none of the pre-whitened time series has AR(1) with $p < 0.05$.

Nonetheless, further analysis of the detrended WHI time series reveals some statistically-significant autocorrelations at lags 2 or higher at six sites where the MKT reveals trends with $p < 0.05$. These sites are: the Cowlitz, Michipicoten, Montreal (Lake Superior), Obey, Sacandaga and Tallapoosa rivers (Table R2). Thus the section on the impacts of serial correlation on the trend analysis in the Supplementary Information (lines 290-296) now includes details of the sites where long-term persistence may play a role on the significance of the trend results. A paragraph has been added to address this issue as follows:

“Several studies⁵⁻⁸ suggest that long-term persistence (beyond lag-1 autocorrelation) may also lead to overestimation of trend significance in hydrometeorological variables. Further analysis reveals that only six sites with statistically-significant trends (two positive and four negative) also exhibit autocorrelations with lag-2 or higher with $p < 0.05$ in their detrended WHI time series. Thus care is required when interpreting the significance of the trends for the Cowlitz, Michipicoten, Montreal (Lake Superior), Obey, Sacandaga and Tallapoosa rivers given the presence of long-term persistence in their WHI time series.”

Table R2: Six sites where the Mann-Kendall test reveals statistically-significant trends in WHI in the presence of statistically-significant autocorrelations (lag 2 or higher) in detrended WHI time series, 1980-2019.

Site	WHI Trend Magnitude (year ⁻¹)	p -value
Cowlitz	5.09×10^{-2}	8.84×10^{-5}
Michipicoten	-7.40×10^{-2}	1.42×10^{-3}
Montreal (Lake Superior)	-4.33×10^{-2}	7.58×10^{-3}
Obey	4.70×10^{-2}	1.67×10^{-2}
Sacandaga	-5.40×10^{-2}	4.59×10^{-3}
Tallapoosa	-3.24×10^{-2}	7.76×10^{-3}

Building on the work of Dinpashoh et al. (2014), Khaliq et al. (2009), Kumar et al. (2009) and Zamani et al. (2017), we plan to explore the role of long-term persistence on WHI trend analyses in a future effort; however, this preliminary work suggests only a few sites in our database exhibit long-term persistence that would impact the significance of the detected WHI trends. Indeed, even if consideration of long-term persistence yielded insignificant trends at all six sites listed in Table R2, we would retain 26 positive and 134 negative significant trends in our database of 479 sites with $n_y \geq 30$ years across the USA and Canada for 1980-2019.

- The MK method does not detect breaks in mean values nor discriminate between sharp and gradual variance values. A statistical test that allows objective detection of breaks in mean values of the analyzed series (e.g., Pettitt, Lombard test) must be used, making it possible to determine rigorously the dates of shifts in mean values of the WHI index.

Correct, the MKT only distinguishes a linear, monotonic trend irrespective whether it arises from an abrupt or a gradual change in WHI. It is possible that some of the reported

trends in WHI arise from a sudden change in operation, the commissioning of a new, or the decommissioning of an old, hydropower facility.

Figure R8: Time series of the WHI for (a) the Churchill River at Churchill Falls Powerhouse, Labrador, (b) Cowlitz River, (c) Michipicoten River, and (d) South Saskatchewan River, 1980-2019. Horizontal blue and red lines identify mean WHI values before and after, respectively, the detection of a statistically-significant change point through the Pettitt test. Note y-axis scales vary between panels.

In response to this comment, we applied the Pettitt (1979) test to verify if any of the statistically-significant linear trends are associated with break points in the WHI time series. There are indeed 109 sites for which statistically-significant break points are detected by the Pettitt test at sites with MKT trends with $p < 0.05$ and at least 30 years of available data during 1980-2019. Figure R8 provides examples of the results of the Pettitt test applied to two sites with statistically-significant positive trends in WHI (Cowlitz and South Saskatchewan rivers) and two sites with statistically-significant negative trends in WHI (Churchill and Michipicoten rivers). This illustrates that the Pettitt test accurately detects years when abrupt changes in WHI appear at all four sites.

For completeness, we have prepared an extra supplementary table with the results of the Pettitt test at all sites with no less than 30 years of available data during 1980-2019. Specifically, Supplementary Table 4 contains results of the Pettitt test statistic U^* , the

corresponding p-value, and the years when a change point is identified. Additionally, the table lists the mean WHI prior to and after the change points. We caution, however, that not all results of the Pettitt test are statistically-significant at the $p < 0.05$ level and must be interpreted with care. In this application, the Pettitt test reports only the most significant break point in a time series that can otherwise include several change points or none at all. Where insufficient data ($n_y < 30$ years) are available to perform the Pettitt test, “NA” is shown in the table for the test statistics.

Thus application of the Pettitt test is helpful in determining rigorously the inception of new management practices in regulated waterways, commissioning or decommissioning of hydropower infrastructure, among other factors that may induce abrupt changes in WHI. A statement on the results of the Pettitt test has been added to the discussion of the MKT results (lines 240-242). We thank Reviewer #3 for this useful comment that provides added value to the WHI trend results.

- It would be useful to apply a classification method (e.g., bottom-up hierarchical classification) to WHI index values to subdivide the 400 sites in classes to improve their characterisation and description.

Thank you for this suggestion to implement a bottom-up hierarchical classification scheme to the WHI results. Following this suggestion, we implemented the “cluster” package in R and used the “agnes” function to perform agglomerative clustering starting with the 1980-2019 mean WHI values at all 500 sites. Aside from the WHI, the data table contains the relevant metadata for each site: latitude, longitude, gauged area, and mean annual discharge. Given the latter two variables span several orders of magnitude, their base 10 logarithms are instead used in the data table. The data are then standardized prior to the cluster analysis with the “agnes” function. Figure R9 provides a dendrogram of the results from the classification method.

Figure R9: Dendrogram illustrating the result of a bottom-up hierarchical classification scheme applied to all 500 sites based on the 1980-2019 mean WHI, coordinates, and the base 10 logarithms of gauged area and mean annual discharge.

While this is an interesting idea, we do not find that the bottom-up hierarchical classification of the WHI provides useful interpretation of our results, particularly in addressing the study's objectives of identifying recent trends in weekly hydropeaking cycles across the USA and Canada. As such, we exclude the results of the bottom-up hierarchical classification from this manuscript but will consider incorporating this in a future effort.

2. Results

Table 1 should be replaced with a table showing the result of all values of the WHI index derived for the 400 sites. See an example

Indices Number of sites % of sites in USA (%) Sites in Canada (%)

>3.5

3.0 – 3.5

2.5 – 3.0

2.0 – 2.5

1.5 – 2.0

1.0 1.5

0.5 – 1.0

0.0 – 0.5

-0.5 – 0.0

-1.0 - -0.5

-1.5 - -1.0

-2.0 - -1.5

-2.5 - -2.0

-3.0 - -2.5

-3.5 - -3.0

< -3.5

N.B. It may be preferable to replace this table by a figure.

In response to this comment we have replaced Table 1 with the binned distribution of WHI values for the USA, Canada and all sites over 1980-2019 (reproduced below as Table R3). The bins used, however, are slightly different from those suggested by Reviewer #3. Instead, they follow the same bin sizes (in increments of 0.75) as those used in the spatial plots. Results in this table are now discussed on lines 139-142 of the paper.

Table R3. Number and percentage of sites in 10 WHI bins in increments of 0.75 for all sites, the USA, and Canada, 1980-2019. WHI bins follow those used in Figure 1 in the paper.

WHI Bin	Sites – All	Sites – All (%)	Sites – USA	Sites – USA (%)	Sites – Canada	Sites – Canada (%)
< -3.00	2	0.4	0	0.0	2	1.3
-3.00 to -2.25	9	1.8	0	0.0	9	5.9
-2.25 to -1.50	22	4.4	2	0.6	20	13.2
-1.50 to -0.75	48	9.6	23	6.6	25	16.4
-0.75 to 0.00	151	30.2	113	32.5	38	25.0
0.00 to 0.75	140	28.0	118	33.9	22	14.5
0.75 to 1.50	65	13.0	46	13.2	19	12.5
1.50 to 2.25	36	7.2	26	7.5	10	6.6
2.25 to 3.00	21	4.2	14	4.0	7	4.6
> 3.00	6	1.2	6	1.7	0	0.0

The previous Table 1 listing the top 10 WHI scores has been moved to the Supplementary Information document as Supplementary Table 2.

References:

- Auer, N. A. Response of spawning lake sturgeons to change in hydroelectric facility operation. *T. Am. Fish. Soc.* **125**, 66-77 (1996).
- Dinpashoh, Y., Mirabbasi, R., Jhajharia, D., Abianeh, H. Z. & Mostafaeipour, A. Effect of short-term and long-term persistence on identification of temporal trends. *J. Hydrol. Eng.* **19**, 617-625 (2014).
- Ferrazzi, M., Woods, R. A. & Botter, G. Climatic signatures in regulated flow regimes across the Central and Eastern United States. *J. Hydrol. Reg. Studies*, **35**, 100809 (2021).
- Khaliq, M. N., Ouarda, T. B. M. J. & Gachon, P. Identification of temporal trends in annual and seasonal low flows occurring in Canadian rivers: the effect of short- and long-term persistence. *J. Hydrol.* **369**, 183-197 (2009).
- Kumar, S., Merwade, V., Kam, J. & Thurner, K. Streamflow trends in Indiana: Effects of long term persistence, precipitation and subsurface drains. *J. Hydrol.* **374**, 171-183 (2009).
- Pettitt, A. N. A non-parametric approach to the change point problem. *J. Appl. Stat.* **28**, 126-135 (1979).
- U.S. Energy Information Administration: <http://iea.org> (2021).
- Yue, S., Pilon, P., Phinney, B. & Cavadias, G. The influence of autocorrelation on the ability to detect trend in hydrological series. *Hydrol. Process.* **16**, 1807-1829 (2002).
- Zamani, R., Mirabbasi, R., Abdollahi, S. & Jhajharia, D. Streamflow trend analysis by considering autocorrelation structure, long-term persistence, and Hurst coefficient in a semi-arid region of Iran, *Theor. Appl. Climatol.* **129**, 33-45 (2017).

Reviewers' Comments:

Reviewer #1:

Remarks to the Author:

The manuscript has been improved. Below are just a handful of editorial recommendations for inaccurate statements and which do not require re-review. The paper is great and impactful as is, and is promising to be cited a lot for "the decreasing in weekly hydropeaking" finding. Without further synthesis however it cannot be cited for any potential reason or regional implications. It seems a bit limited for a Nature paper.

The authors maintained an analysis throughout the now-500 locations with a rich discussion on the potential reasons for reduction in hydropeaking without further investigating support for those inferences. I recommend authors to engage in a limited synthesis (because there is already a lot in the paper) to enhance the impact of the paper. With or without this additional synthesis, I strongly recommend to further work on figure 4 for highlighting and clarifying the synthesis of the finding. Suggestions are also provided below.

1) First paragraph of results section. "While many dams in North America have multiple purposes, hydropower generation remains a principal function" . That is not the case throughout the western USA especially for federal dams where hydropower is the purpose with the least priority. I would remove this statement or rephrase to " Most dams are operated for multi-purposes shaping seasonal and subseasonal patterns. Hydropower remains a principal component for sub-monthly variations along with flood control."

2) "Clusters of negative WHI trends lie primarily within the Western Electricity Coordinating Council, Northeast Power Coordinating Council and SERC Reliability Corporation power grid interconnections". The authors refer to "reliability councils in charge of resource adequacy and arbitrage" when in fact they should refer to the actual interconnects "Western, Northeastern and Southeastern Interconnects".

3) "Recent cost reductions for the production of solar and wind energy may also lead to new types of services offered by hydropower such as capacity markets. Furthermore, the rapid 328 increase in electricity production from non-hydro renewable sources coincides with the sharp decline of weekly hydropeaking intensity in the 2010s (Supplementary Fig. 12)." The two sentences do not flow together. The second sentence would make more sense coming first, leading to hydro providing new services. I would not mention the cost of wind and solar as those are high subsidized in the US and considered as must-take by the grid.

4) "Alternatively, wet years may lead utilities to generate continuous baseload energy instead of peaking hydropower, inducing a similar effect. The relatively wet climate of the 2010s could account for part of the recent declines in WHI across the USA and Canada." Is there a reference to back up a wet year from a hydropower perspective? 2014-16 was the worse drought in California and 2015 a severe drought in the Northwest, and 2011 the worse drought of record in Texas for water-dependent electricity generation.

5) Figures: figure 4 is very much around showing the data while panels k and l are the most informative supporting the synthesis in the paper. Panel l is very hard to understand without the color legend on the side. The caption says that the week starts on FS while other graphs start on SS. This is a bit inconsistent which contributes the overall lack of clarity for this panel. I understand that authors maintain all the panels, k and l are however those going to be picked up by media and other authors to build on this work. They deserve to be highlighted and more self standing.

6) the overall analysis still focuses on the WHI value at individual locations. Regulation and operations

(governance) are however at the watershed scale and interconnect scale. The paper has a rich discussion which remains based on inference while using panels k and i from figure 4 by interconnect for example (western US, western Canada, ERCOT, Southeastern, MISO, etc) would provide more support on the discussion based on the generation portfolio. Similarly hydrologic regions could be used as well. The manuscript has an impactful message but lacks a synthesis to increase the impact of the paper.

Reviewer #2:

Remarks to the Author:

I'd like to thank the authors for fully addressing my comments and improving the manuscript accordingly. The analyses of the WHI temporal evolutions have offered some new insights. The authors have also done a great work investigating the relationship between non-hydropower electricity and WHI. I do not have additional comments for the revised manuscript.

Reviewer #3:

Remarks to the Author:

Dear Editor,

I have carefully read the answers to the questions raised by the three reviewers and the corrections made by the authors to their manuscript. As far as I'm concerned, I'm happy with the changes made by the authors. I recommend acceptance of the article in its current revised form.

Best Regards,
Prof. Assani

RESPONSE DOCUMENT

NCOMMS-21-12638A

REVIEWER COMMENTS

Reviewer #1 (Remarks to the Author):

The manuscript has been improved. Below are just a handful of editorial recommendations for inaccurate statements and which do not require re-review. The paper is great and impactful as is, and is promising to be cited a lot for "the decreasing in weekly hydropeaking" finding. Without further synthesis however it cannot be cited for any potential reason or regional implications. It seems a bit limited for a Nature paper.

The authors maintained an analysis throughout the now-500 locations with a rich discussion on the potential reasons for reduction in hydropeaking without further investigating support for those inferences. I recommend authors to engage in a limited synthesis (because there is already a lot in the paper) to enhance the impact of the paper. With or without this additional synthesis, I strongly recommend to further work on figure 4 for highlighting and clarifying the synthesis of the finding. Suggestions are also provided below.

We sincerely thank Reviewer #1 for the additional constructive comments provided in this report on our paper. We address each of Reviewer #1's comments with a point-by-point response in bold lettering. As outlined below, a limited synthesis of the results is now provided in our final remarks. We anticipate reporting further details (including analyses at different spatio-temporal scales) and synthesizing our results in future publications. Thank you for taking the time to carefully read and review our revised manuscript as we feel that by addressing these comments our paper is now even stronger.

1) First paragraph of results section. "While many dams in North America have multiple purposes, hydropower generation remains a principal function". That is not the case throughout the western USA especially for federal dams where hydropower is the purpose with the least priority. I would remove this statement or rephrase to "Most dams are operated for multi-purposes shaping seasonal and subseasonal patterns. Hydropower remains a principal component for sub-monthly variations along with flood control."

We agree with this statement and have modified the text accordingly. Lines 100-102 now state as follows: "Most dams in North America are operated for multi-purposes shaping seasonal and subseasonal patterns. Hydropower remains a principal component for sub-monthly variations along with flood control."

2) "Clusters of negative WHI trends lie primarily within the Western Electricity Coordinating Council, Northeast Power Coordinating Council and SERC Reliability Corporation power grid interconnections". The authors refer to "reliability councils in charge of resource adequacy and arbitrage" when in fact they should refer to the actual interconnects "Western, Northeastern and

Southeastern Interconnects".

As per Reviewer #1's suggestion, the text on lines 236-237 has been modified to specify that the clusters of negative WHI lie within the Western, Northeastern and Southeastern Interconnects.

3) "Recent cost reductions for the production of solar and wind energy may also lead to new types of services offered by hydropower such as capacity markets. Furthermore, the rapid increase in electricity production from non-hydro renewable sources coincides with the sharp decline of weekly hydropeaking intensity in the 2010s (Supplementary Fig. 12)." The two sentences do not flow together. The second sentence would make more sense coming first, leading to hydro providing new services. I would not mention the cost of wind and solar as those are high subsidized in the US and considered as must-take by the grid.

We concur with this comment and have updated the text on lines 314-319 as follows: "Solar and wind energy production activate during favourable weather conditions with hydropower otherwise matching the demand, which may disrupt the typical weekly pattern in regulated flows while allowing hydropower to offer new types of services such as capacity markets. Furthermore, the rapid increase in electricity production from non-hydro renewable sources coincides with the sharp decline of weekly hydropeaking intensity in the 2010s (Supplementary Fig. 12)."

4) "Alternatively, wet years may lead utilities to generate continuous baseload energy instead of peaking hydropower, inducing a similar effect. The relatively wet climate of the 2010s could account for part of the recent declines in WHI across the USA and Canada." Is there a reference to back up a wet year from a hydropower perspective? 2014-16 was the worse drought in California and 2015 a severe drought in the Northwest, and 2011 the worse drought of record in Texas for water-dependent electricity generation.

The statement that 2010-2019 was a relatively wet decade is based on the spatial distribution of the decadal standardized discharge anomalies reported in Supplemental Figure 8b and reproduced below (Figure R1). Based on streamflow data, this illustrates the 2010s was generally a wet decade relative to others between the 1920s to 2000s across the northern two-thirds of the study area. This plot also confirms the Reviewer's remark in regards to the prolonged drought conditions in the southwestern and southern USA and parts of the Pacific Northwest (Oregon, Idaho) yielding negative discharge anomalies. The southeastern USA (south of Tennessee and North Carolina) also shows predominantly negative standardized discharge anomalies. For the remainder of the study area including the northeastern and north-central USA and Canada, most discharge anomalies are positive, indicating generally wet conditions in the 2010s. This pattern is consistent with recent precipitation anomalies reported across the USA (e.g. <https://www.climate.gov/media/13465>).

Out of the 500 study sites with available data in the 2010s, 67% showed positive discharge anomalies while 33% reported negative anomalies. Thus the statement about the 2010s

being a relatively wet decade applies only to two-thirds of the study domain. Thus we have adjusted the text on lines 338-339 to reflect this, as follows:

“The relatively wet climate of the 2010s could account for part of the recent declines in WHI across Canada and the northern half of the conterminous USA.”

Figure R1: Spatial distribution of decadal standardized discharge anomalies at 500 sites across the USA and Canada, 2010-2019. Negative values indicate relatively dry conditions while positive values denote relatively wet conditions.

5) Figures: figure 4 is very much around showing the data while panels k and l are the most informative supporting the synthesis in the paper. Panel l is very hard to understand without the color legend on the side. The caption says that the week starts on FS while other graphs start on SS. This is a bit inconsistent which contributes the overall lack of clarity for this panel. I understand that authors maintain all the panels, k and l are however those going to be picked up by media and other authors to build on this work. They deserve to be highlighted and more self standing.

Thank you for these thoughtful suggestions. In response to this comment, we have transferred panels k and l originally in Figure 4 into a separate figure (a new Figure 5) so

that they stand alone and better highlight a synthesis of our results. Color legends have been added to both plots but the order of the two days of the week has not been modified in the final panel – this is to retain the same vertical layout in the presentation of the results in the bar graph (new Figure 5b) with the legends used in Figure 4. The addition of a legend in this panel should hopefully eliminate any issue with the interpretation of these results. The associated text (lines 214-219) describing these results has also been shifted to a separate paragraph so that these findings stand out better in our paper.

6) the overall analysis still focuses on the WHI value at individual locations. Regulation and operations (governance) are however at the watershed scale and interconnect scale. The paper has a rich discussion which remains based on inference while using panels k and i from figure 4 by interconnect for example (western US, western canada, ercot, southeastern, MISO, etc) would provide more support on the discussion based on the generation portfolio. Similarly hydrologic regions could be used as well. The manuscript has an impactful message but lacks a synthesis to increase the impact of the paper.

We concur that a comprehensive synthesis of the results is lacking in the paper; however, the strict length limitations imposed by *Nature Communications* prevents us from adding a new section to synthesize our main results. Nevertheless, in response to this comment, we have added a limited synthesis with the following sentences (lines 453-461) in the final part of the Discussion section now titled “Summary and synthesis”:

Our analyses reveal that 29% of sites with at least three decades of available data during 1980-2019 exhibit locally statistically-significant declines in WHI while only 6% show inclines. Moreover, the fraction of sites with $WHI \geq 1.5$ dropped by half from the 2000s to the 2010s reverting to a value observed in the 1920s. Major watersheds observing significant declines in weekly hydropeaking include the Alabama, Columbia, Cumberland, Great Lakes-St. Lawrence, and upper Mississippi, which fall within the Eastern and Western Interconnects. Regional clusters of declining WHI highlight hydropower operations and river regulation governed at the watershed-, interconnect- and utility-scale.

We anticipate following up on this study with a more profound regional analysis of WHI trends including at the interconnect- and watershed-scales, and perhaps even at the utility scale, if relevant electricity generation data can be acquired for analysis. Thus we now conclude our paper with the following statement (lines 471-473):

Lastly, detailed investigations at various spatial (e.g., watershed, interconnect, utility) and temporal (e.g., seasonal) scales should be undertaken to elucidate the role of governing agencies and hydroclimate on hydropeaking globally.

Sincere thanks to Reviewer #1 for providing supplemental remarks on our revised manuscript that continue to improve the presentation and interpretation of our results.

Reviewer #2 (Remarks to the Author):

I'd like to thank the authors for fully addressing my comments and improving the manuscript accordingly. The analyses of the WHI temporal evolutions have offered some new insights. The authors have also done a great work investigating the relationship between non-hydropower electricity and WHI. I do not have additional comments for the revised manuscript.

Our sincere thanks to Reviewer #2 for this positive overview of our revised manuscript and for stating that our revisions have fully addressed the Reviewer's earlier comments. We sincerely appreciate the Reviewer's time and effort in assessing the overall revised manuscript and response document.

Reviewer #3 (Remarks to the Author):

Dear Editor,

I have carefully read the answers to the questions raised by the three reviewers and the corrections made by the authors to their manuscript. As far as I'm concern, I'm happy with the changes made by the authors. I recommend acceptance of the article in its current revised form.

Best Regards,
Prof. Assani

Sincere thanks to Reviewer #3 for a positive overview of our revised manuscript and that the earlier comments were fully addressed in our revisions. We express our deep gratitude to the Reviewer for the time and effort placed into assessing our revised manuscript and our responses to the comments we received from the three referees.